# CHD-associated enhancers shape human cardiomyocyte lineage commitment

Daniel A Armendariz[1†], Sean C Goetsch[2†], Anjana Sundarrajan[1],
Sushama Sivakumar[2], Yihan Wang[1], Shiqi Xie[1‡], Nikhil V Munshi[2,3,4]*,
Gary C Hon[1,4,5]*

[1]Cecil H. and Ida Green Center for Reproductive Biology Sciences, University of Texas Southwestern Medical Center, Dallas, United States; [2]Department of Internal Medicine, University of Texas Southwestern Medical Center, Dallas, United States; [3]Division of Cardiology, Department of Molecular Biology, McDermott Center for Human Growth and Development, University of Texas Southwestern Medical Center, Dallas, United States; [4]Hamon Center for Regenerative Science and Medicine, University of Texas Southwestern Medical Center, Dallas, United States; [5]Lyda Hill Department of Bioinformatics, Department of Obstetrics and Gynecology, University of Texas Southwestern Medical Center, Dallas, United States

*For correspondence:
Nikhil.Munshi@UTSouthwestern.
edu (NVM);
Gary.Hon@UTSouthwestern.edu
(GCH)

†These authors contributed equally to this work

Present address: ‡Genentech, 1 DNA Way, South San Francisco, United States

**Abstract** Enhancers orchestrate gene expression programs that drive multicellular development and lineage commitment. Thus, genetic variants at enhancers are thought to contribute to developmental diseases by altering cell fate commitment. However, while many variant-containing enhancers have been identified, studies to endogenously test the impact of these enhancers on lineage commitment have been lacking. We perform a single-cell CRISPRi screen to assess the endogenous roles of 25 enhancers and putative cardiac target genes implicated in genetic studies of congenital heart defects (CHDs). We identify 16 enhancers whose repression leads to deficient differentiation of human cardiomyocytes (CMs). A focused CRISPRi validation screen shows that repression of TBX5 enhancers delays the transcriptional switch from mid- to late-stage CM states. Endogenous genetic deletions of two TBX5 enhancers phenocopy epigenetic perturbations. Together, these results identify critical enhancers of cardiac development and suggest that misregulation of these enhancers could contribute to cardiac defects in human patients.

## Editor's evaluation

The work presented is a valuable assessment of a broad set of regulatory elements that coordinate cardiac differentiation. The approach is broadly applicable, and the results point to important mechanisms of gene regulation and differentiation, with implications for future studies in non-coding variation.

## Introduction

Congenital heart defects (CHDs) encompass a broad range of cardiac malformations that impact 1% of all births (*Hoffman and Kaplan, 2002*). However, the genetic basis for 54% of familial and 80% of sporadic CHD cases remain unknown (*Blue et al., 2017*). Genome-wide association studies and whole genome sequencing of CHD patients have identified thousands of variants that associate with CHD (*Agopian et al., 2017*; *Cordell et al., 2013*; *Richter et al., 2020*). Most efforts have focused on functionally characterizing coding variants. For example, traditional linkage analysis in patients with

Holt-Oram syndrome implicated the gene for the TBX5 transcription factor, which harbors variants predicted to disrupt protein structure and/or function in affected patients (*Basson et al., 1997*).

TBX5 is a key regulator of cardiac development, which requires precise temporal expression: early expression of TBX5 is necessary for CM maturation and later expression is required for specification of the cardiac conduction system (*Bruneau et al., 2001*; *Steimle and Moskowitz, 2017*). However, protein-coding variants account for only a small fraction of CHD risk. In contrast, the vast majority of de novo variants are non-coding and potentially modify the activity of transcriptional enhancers (*Richter et al., 2020*). Only a handful have been functionally examined, including a non-coding CHD-associated variant which has been shown to impair proper TBX5 expression (*Smemo et al., 2012*). Thus, there remains a critical gap in knowledge between CHD-associated non-coding regions and their contribution to CHD.

Recent studies have explored high-throughput genomic approaches to assess the roles of enhancers and regulatory variants. For example, *Richter et al., 2020* performed whole genome sequencing in CHD patients to identify thousands of non-coding variants. Following extensive computational filtering to narrow the list of putative causal mutations, the authors used massively parallel reporter assays to functionally identify five CHD-associated variants that impact enhancer activity. However, reporter assays do not assess how variants function in the endogenous genomic context. CRISPR screens have been developed to identify regulatory elements that contribute to screenable phenotypes following epigenetic silencing or genetic deletion (*Fulco et al., 2016*). While powerful, this strategy has focused on screenable phenotypes in homogeneous cell populations. Thus, traditional CRISPR screens would lose information about how enhancers regulate specification of distinct cell types during development.

To address this deficiency, single-cell approaches have been applied in isogenic systems to delineate how perturbations affect lineage commitment. For example, *Kathiriya et al., 2021* generated an allelic series of TBX5 locus modifications in iPSCs followed by CM differentiation and single-cell sequencing to show that TBX5 exhibits a dosage-sensitive effect altering the trajectory of CM specification. Recent studies have extended this approach using single-cell CRISPR screens to systematically test the roles of key genes in endoderm and neuronal specification (*Genga et al., 2019*; *Tian et al., 2019*). Similar approaches have also been applied for high-throughput examination of enhancer functions in stable, homogeneous cell lines like K562 cells (*Gasperini et al., 2019*; *Xie et al., 2017*). A combination of these approaches will likely be required to systematically study the endogenous functions of CHD-associated enhancers during CM lineage commitment.

Here, we apply a single-cell CRISPRi screen for 25 CHD-associated enhancers in a model of human embryonic stem cell (hESC) to CM differentiation. We identify a subset of 16 enhancers which, when perturbed, results in deficient CM specification. In focused validation studies, we show that CRISPRi repression or genetic deletion of TBX5 enhancers leads to dysregulation of the cardiac gene expression program, enrichment of early CM cell states, and a depletion of mature CM cell states.

## Results

### Single-cell screens of CHD-associated enhancers during CM differentiation

Whole genome sequencing has identified >50,000 de novo variants in 749 CHD patients (*Richter et al., 2020*). Some of these variants may contribute to CHD by modulating the activity of transcriptional enhancers that are active in cardiac development. To identify these enhancers for functional study, we overlapped variants with published epigenetic datasets of active enhancers including open chromatin (ATAC-Seq) and histone acetylation (H3K27ac) (*Liu et al., 2017*; *Zhang et al., 2019*). We prioritized a diverse set of CHDs from atrial and septal defects to tetralogy of Fallot (*Supplementary file 1*). We also filtered for enhancers within 100 kb of genes with known roles in heart development. Overall, we prioritized 25 enhancers, including those in proximity to known cardiac regulators, with many exhibiting high predicted sequence conservation (*Siepel et al., 2005*; *van den Hoogenhof et al., 2018*; *Figure 1A*). Epigenetic evidence indicates that many of these enhancers and their putative downstream genes become active at early time points of cardiac differentiation (*Tompkins et al., 2016*; *Figure 1B–C*). Overall, these data suggest that the prioritized CHD-associated enhancers may

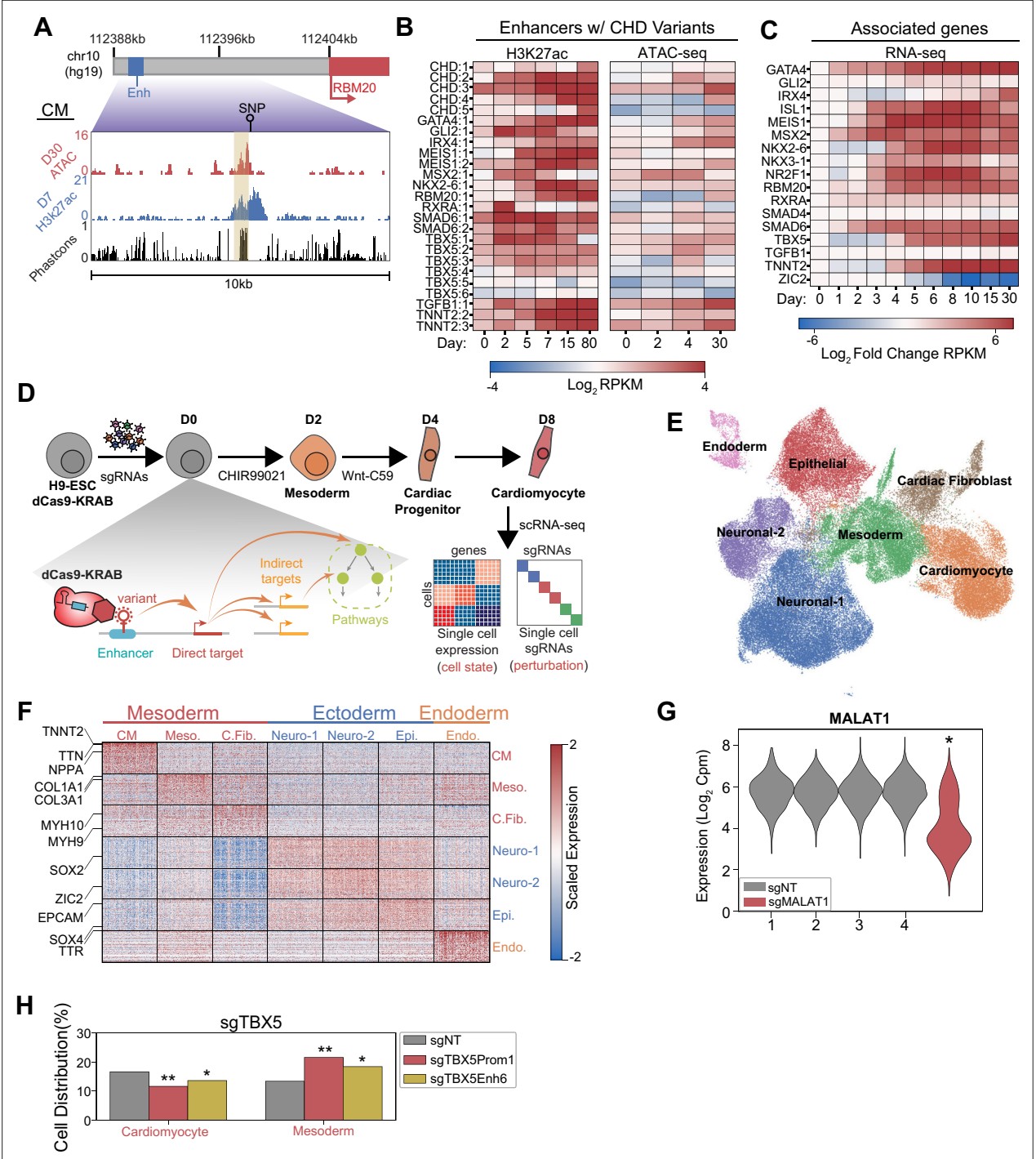

**Figure 1.** Single-cell screens of congenital heart defect (CHD)-associated enhancers during cardiomyocyte (CM) differentiation. (**A**) Genome browser snapshot emphasizing features of a targeted enhancer about 15 kb upstream of the RBM20 gene. Yellow highlights enhancer region. (**B**) H3K27ac and open chromatin (ATAC-Seq) enrichment for targeted enhancers across multiple time points of CM differentiation (*Liu et al., 2017*; *Tompkins et al., 2016*; *Zhang et al., 2019*). (**C**) Expression of putative target genes across multiple stages of CM differentiation. Expression defined as fold change over the day 0 expression of a target gene (*Tompkins et al., 2016*). (**D**) Schematic of single-cell CRISPRi screen. H9-dCas9-KRAB cells are infected with a lentiviral sgRNA library and differentiated over 8 days into CMs followed by scRNA-seq. Individual cells are linked to sgRNA perturbations and changes in transcriptional cell state. (**E**) UMAP visualization of H9-derived cells after 8 days of differentiation. Seven Louvain clusters indicated. (**F**) Expression of the top 100 cluster defining genes for each Louvain cluster cell type. (**G**) MALAT1 expression in control (non-targeting) and sgMALAT1 cells (*p<0.05 by Mann-Whitney U). (**H**) Distribution of cells receiving sgNT, sgTBX5 PROM1, sgTBX5 ENH6 that differentiate into CM and mesoderm states (*p<0.05 and **p<0.001 by hypergeometric test).

*Figure 1 continued on next page*

*Figure 1 continued*

The online version of this article includes the following figure supplement(s) for figure 1:

**Figure supplement 1.** Single-cell CRISPRi screen validation and statistics.

have roles in early cardiac lineage commitment, and we hypothesize that perturbations of these enhancers may impact cell fate.

To test this hypothesis, we used CRISPRi-mediated single-cell perturbation screens to assess the functions of CHD-associated enhancers during early cardiac lineage commitment (*Figure 1—figure supplement 1A–B*). We modeled this process by differentiating hESCs toward CMs through WNT modulation. Notably, to preserve cellular heterogeneity, we performed CM differentiation without metabolic selection (*Tohyama et al., 2013*). To capture the early events of lineage commitment, we examined cells at day 8 of differentiation (*Figure 1D*). Overall, we targeted 397 sgRNAs spanning 25 enhancers, 19 promoters of their putative target genes, and non-genome targeting (NT) controls (*Supplementary file 1*). After single-cell RNA sequencing (RNA-seq) and stringent filtering to remove low-quality cells and doublets, we retained 80,343 high-quality cells for downstream analysis. On average, we detected one sgRNA in each cell (*Figure 1—figure supplement 1C–E* and *Supplementary file 2*).

To define the cell populations during early differentiation, we clustered cells to identify seven distinct cell populations across all three germ layers (*Figure 1E* and *Supplementary file 3*). We focused on the three mesoderm-derived clusters due to their relevance to cardiac development. These clusters included TNNT2+ CMs, FN1+ mesodermal cells (*Cheng et al., 2013*), and COL3A1+ cardiac fibroblasts (*van Nieuwenhoven et al., 2013*; *Figure 1F* and *Figure 1—figure supplement 1F*). Verifying the efficiency of CRISPRi-mediated repression, we observed robust depletion of control genes for positive control sgRNAs ($p_{Malat1}$ <2.2e-308, Mann-Whitney U) (*Figure 1G* and *Figure 1—figure supplement 1G*).

## CRISPRi of CHD-associated enhancers delays CM differentiation

To test the ability of our approach to detect changes in cell state from CRISPRi perturbations, we included positive control sgRNAs targeting the promoter of well-known cardiac genes as an internal quality control measure. We first examined sgRNAs targeting the promoter of ZIC2, a known regulator of both neuronal and cardiac lineages necessary for the successful differentiation of both (*Luo et al., 2015*; *Xu et al., 2020*). Compared to cells with non-target sgRNAs, the cells containing sgRNAs for ZIC2 promoter were significantly depleted for neuronal and cardiac populations ($p_{neuronal}$ = 4.7e-61; $p_{cardiomyocytes}$ = 1.9e-5, hypergeometric test). Consistent with a stalled differentiation state, these cells exhibit enrichment for an early mesodermal state (*Figure 1—figure supplement 1H*). Second, based on published knockout studies showing that TBX5 repression prevents proper CM specification (*Kathiriya et al., 2021*), we also included sgRNAs targeting the promoter of TBX5. We observed that loss of TBX5 expression led to a depletion in CMs ($p_{TBX5\ promoter}$ = 2.0e-5; $p_{TBX5\ enhancer\ 6}$ = 0.025, hypergeometric test) and a corresponding increase in early mesoderm cells ($p_{TBX5\ promoter}$ = 3.4e-11; $p_{TBX5\ Enhancer\ 6}$ = 4.1e-4, hypergeometric test) (*Figure 1H*). Interestingly, perturbation of a CHD-associated enhancer of TBX5 phenocopies perturbation of the promoter in terms of cell state changes. Bulk qPCR analysis confirms that CRISPRi of TBX5 enhancers results in loss of TBX5 expression (*Figure 1—figure supplement 1I*). Taken together, this analysis demonstrates that CRISPRi repression at the TBX5 promoter functions as expected during CM differentiation and suggests further that repression of TBX5 expression through enhancer perturbation similarly depletes CM lineage commitment through loss of downstream target expression.

Next, to gain higher resolution insights on how enhancer perturbation influences lineage commitment, we focused on cells in the cardiac lineage (CM, mesoderm). We identified a trajectory of differentiating CMs composed of four distinct subpopulations including SOX4+ progenitors (*Paul et al., 2014*), FN1+ early-stage CMs, ACTA2+ mid-stage CMs (*Potta et al., 2010*), and NPPA+ atrial-like late-stage CMs (*Figure 2A* and *Figure 2—figure supplement 1A*). Consistent with the established process of CM commitment, pseudotime analysis orders these cell states with increasing expression of cardiac maturation genes such as TNNT2 (*Figure 2B–C*). Further validating this trajectory, we observe consistency with published bulk RNA-seq datasets of CM differentiation (*Tompkins et al.,*

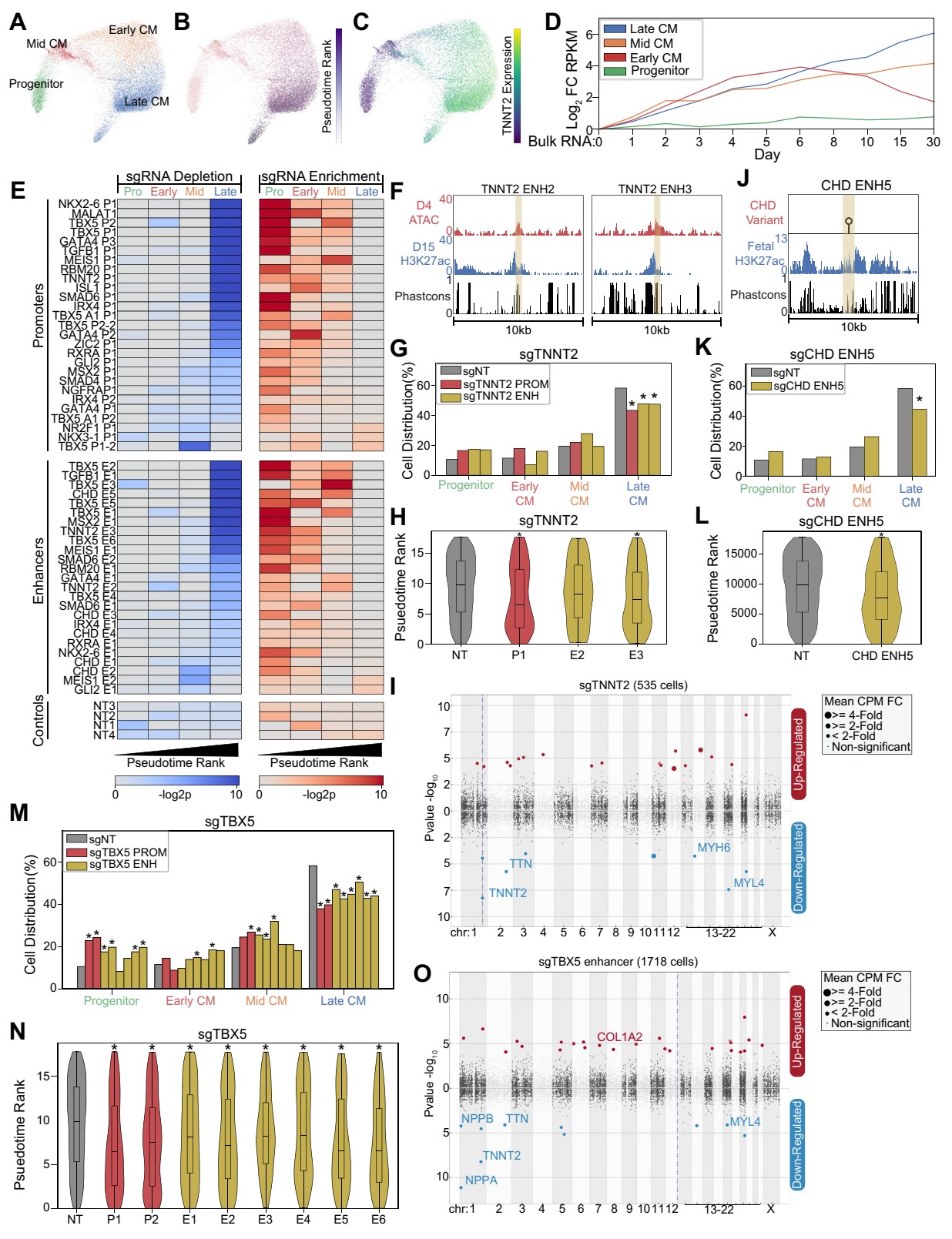

**Figure 2.** CRISPRi of congenital heart defect (CHD)-associated enhancers delays cardiomyocyte (CM) differentiation. (**A**) PHATE visualization of CM cells with four Louvain clusters. (**B**) Feature plot of pseudotime ranking of cells across PHATE trajectory. (**C**) Feature plot of TNNT2 expression. We defined the top 100 genes for each of the four CM subtypes. Shown is the expression of these gene sets in bulk RNA-seq experiments of CM differentiation (*Tompkins et al., 2016*). Expression defined as fold change over day 0. (**E**) Enrichment (right) and depletion (left) of cells with distinct perturbations

Figure 2 continued

across CM subpopulations (p-values: hypergeometric test). Top: Targeted promoters; middle: targeted enhancers; bottom: non-targeting sgRNAs. (F) Genome browser tracks of chromatin and sequence conservation at two putative TNNT2 enhancers (ENH2 and ENH3). Yellow region denotes enhancer boundaries. (G) Distribution of states for cells receiving sgRNAs targeting TNNT2 promoter and enhancers (*p<0.05 by hypergeometric test). (H) Distribution of pseudotime rank for cells receiving sgRNAs targeting TNNT2 promoter and enhancers (*p<0.05 by Mann-Whitney U) ($n_{NT}$ = 983 cells, $n_{P1}$ = 145 cells, $n_{E2}$ = 86 cells, $n_{E3}$ = 311 cells). (I) Differentially expressed genes in CM cells receiving sgRNAs targeting TNNT2 promoter or enhancers. In this Manhattan plot, the horizontal axis indicates genomic coordinates, with the dotted line indicating the targeted TNNT2 promoter. The vertical axis indicates differential expression (p-value), with positive values representing increased expression and negative values representing decreased expression. (J) Genome browser track of fetal human heart H3K27ac for CHD ENH5. Yellow region denotes enhancer boundaries. (K) Distribution of states for cells receiving sgRNAs targeting CHD ENH5 (*p<0.05 by hypergeometric test). (L) Distribution of pseudotime rank for cells receiving sgRNAs targeting CHD ENH5 (*p<0.05 by Mann-Whitney U) ($n_{NT}$ = 983 cells, $n_{E5}$ = 281 cells). (M) Distribution of states for cells receiving sgRNAs targeting TBX5 promoters and enhancers (*p<0.05 by hypergeometric test). (N) Distribution of pseudotime rank for cells receiving sgRNAs targeting TBX5 promoter and enhancers (*p<0.05 by Mann-Whitney U) ($n_{NT}$ = 983 cells, $n_{P1}$ = 179 cells, $n_{P2}$ = 271 cells, $n_{E1}$ = 348 cells, $n_{E2}$ = 466 cells, $n_{E3}$ = 437 cells, $n_{E4}$ = 166 cells, $n_{E5}$ = 205 cells, $n_{E6}$ = 127 cells). (O) Differentially expressed genes in CM cells receiving sgRNAs targeting a TBX5 enhancer (as described in 2I).

The online version of this article includes the following figure supplement(s) for figure 2:

Figure supplement 1. Single-cell CRISPRi screen cardiomyocyte (CM) subpopulation marker gene expression.

2016; Figure 2D). Thus, our single-cell dataset recapitulates the expected patterns of CM cell fate commitment.

A previous study established that loss of cardiogenic TFs deviates cells from a WT trajectory (Kathiriya et al., 2021). To examine the effect of enhancers, we compared the cell state distribution in perturbed cells. Surprisingly, we observed that perturbation of 16 enhancers led to a depletion of late-stage CMs. In contrast, only two enhancers were depleted in earlier CM states. This depletion of late-stage CMs coincides with an enrichment in progenitor, early-CM, and mid-CM states (Figure 2E and Figure 2—figure supplement 1B–C). Since connecting enhancers with their cognate promoters can be challenging, we only considered enhancer perturbations that produced phenotypes similar to paired promoter perturbation. Using this stringent approach, we detected 14 out of 25 enhancer perturbations that caused changes in cell state distribution, although this likely represents an underestimate of the true hit rate. To account for the low expression of many cardiac genes which makes differential gene expression analysis underpowered, we performed targeted sequencing to measure the expression of key cardiac gene targets (Figure 2—figure supplement 1D). This analysis showed further consistency between promoter-targeting and enhancer-targeting sgRNAs (Figure 2—figure supplement 1E).

Next, we illustrate several examples. First, we targeted two evolutionarily conserved enhancers near the CM structural protein TNNT2 that exhibit active enhancer chromatin during CM differentiation (Figure 2F). We observed that CRISPRi repression of either the TNNT2 promoter or one of the two targeted nearby enhancers resulted in a depletion of late-CM cells ($p_{TNNT2\ P1}$ = 0.0005; $p_{TNNT2\ E2}$ = 0.03; $p_{TNNT2\ E3}$ = 0.001, hypergeometric test) (Figure 2G). These perturbations also yielded similar changes in pseudotime cell state along with reduced expression of TNNT2 and other cardiac genes within the CM populations ($p_{TNNT2\ P1}$ = 0.0004; $p_{TNNT2\ E2}$ = 0.07; $p_{TNNT2\ E3}$ = 1.2e-6, Mann-Whitney U) (Figure 2H–I and Supplementary file 4). Second, we also targeted a CHD-associated enhancer with a variant characterized by reporter assay analysis (Gilsbach et al., 2018; Richter et al., 2020; Figure 2J). Repression of this enhancer also impacted lineage specification, depleting cells in the late-CM state (p=2.5e-5, hypergeometric test) (p=3.6e-5,Mann-Whitney U) (Figure 2K–L). Third, we examined how perturbation of six distinct TBX5 enhancers affected CM trajectory. CRISPRi-mediated repression of any of the six TBX5 enhancers led to a depletion of late-CM ($p_{enhancers1-6}$ between 3.7e-2 and 1.9e-8, hypergeometric test) ($p_{enhancers1-6}$ between 2.5e-2 and 6.8e-9, Mann-Whitney U) (Figure 2E and M–N). Cells with sgRNAs targeting TBX5 enhancers showed reduced expression TBX5 downstream targets, including NPPA and NPPB (Figure 2O and Supplementary file 4; Ang et al., 2016; Houweling et al., 2005; Luna-Zurita et al., 2016). Consistently, we also observed changes in cell state upon repression of TBX5 promoters. In sum, we observe that perturbation of CHD-associated enhancers, particularly for TBX5, results in deficient CM differentiation. These results suggest that these enhancers have roles in CM lineage commitment.

## A focused validation screen demonstrates that TBX5 enhancers modulate CM cell fate

To validate the results of the large screen above, we performed a smaller single-cell CRISPRi screen focused on TBX5 regulatory elements. Three of the enhancers (Enh4, Enh5, and Enh6) were previously identified through sequence conservation and exhibited reporter activity (*Smemo et al., 2012*). The remaining three enhancers (Enh1, Enh2, Enh3) were identified in this study based on epigenetic hallmarks of active enhancers (*Figure 3A*). After transduction of control and targeting sgRNAs, CM differentiation, and sequencing, we identified three primary clusters comprising all germ layers (*Figure 3B*, *Figure 3—figure supplement 1A*, and *Supplementary file 3*), with 3102 cells corresponding to CMs based on TNNT2 expression (*Figure 3C*).

Consistent with our initial screen, we observed that cells with TBX5 enhancer repression were significantly depleted in the late-CM state ($p_{enhancers1-6}$ between 1.3e-5 and 1.3e-19, hypergeometric test) (*Figure 3D–E* and *Figure 3—figure supplement 1B*). We also observed activation of ACTA2 and repression of NPPA in cells with perturbed TBX5 enhancers ($p_{ACTA2}$=3.1e-7; $p_{NPPA}$ = 8.2e-24, Mann-Whitney U) (*Figure 3F*, *Figure 3—figure supplement 1C*, and *Supplementary file 4*), consistent with a previous study that knocked out TBX5 (*Kathiriya et al., 2021*).

Next, we performed focused analysis on CM populations to identify three clusters of cell subpopulations: CM progenitors (FN1+), mid-CM (ACTA2+), and late CM (NPPA+) (*Figure 4A* and *Figure 4—figure supplement 1A*). Pseudotime analysis orders these cell populations by increasing expression of known cardiac maturation genes including NPPA and TNNT2 (*Figure 4B–D*). Overall, our focused screen contains 122 cells balanced across TBX5 enhancer perturbations that attain CM states. Since this number is relatively small, we pooled these cells for downstream analyses. Consistent with our larger screen, we observe a significant depletion of late CMs when TBX5 enhancers are repressed, with an increase in mid-CMs (*Figure 4E–F* and *Figure 4—figure supplement 1B*) and a significant decrease in pseudotime rank ($p_{late\ CM}$ = 3.8e-18; $p_{mid-CM}$ = 7.8e-13, hypergeometric test) (p=2.2e-5, Mann-Whitney U) (*Figure 4G*). This coincided with downregulation of known targets of TBX5 (*Figure 4—figure supplement 1C* and *Supplementary file 4*). Cell label transfer of initial screen CM annotations onto focus screen CMs showed consistent clustering and retention of late-CM depletion under TBX5 enhancer knockdown conditions (*Figure 4—figure supplement 1D–E*). Overall, results from our focused validation screen are consistent with the observation that TBX5 enhancer repression causes deficient CM lineage commitment.

## TBX5 enhancer repression alters CM molecular signatures

While TBX5 enhancer perturbation results in a reduction of late-CM cells, we observe that a small number of cells still reach the state. We next asked whether these cells exhibit altered molecular signatures. First, focusing on the late-CM state, we defined 80 genes that are specifically expressed in NT control cells. These genes coincide with expected CM markers, such as NPPA and TTN, which continuously increase in expression throughout CM differentiation. Interestingly, we find that this set of late-CM genes is depleted in the subset of cells with perturbed TBX5 enhancers (denoted as sgTBX5enh) that reach the late-CM state (p=1.7e-3, Z-test) (*Figure 4H–I* and *Supplementary file 5*). Several notable cardiac genes including NPPA and NPPB exhibit pronounced repression in perturbed cells (*Figure 4J–K*). These observations suggest that, although some sgTBX5Enh cells can reach a late-CM cell state, these cells have deficient activation of late-CM genes. We next asked if these cells also harbored aberrant expression of mid-CM cell signatures.

We identified a set of 18 genes that define the mid-CM state, which includes the structural proteins MYL4 and MYL6. Pseudotime analysis shows that NT control cells repress mid-CM genes as they enter the late-CM stage (*Figure 4L*). In contrast, we observe that sgTBX5Enh cells retain higher expression levels of mid-CM genes in the late-CM state (p=2.6e-4, Z-test) (*Figure 4M* and *Supplementary file 5*). For example, perturbed cells have a 48% increase in HAS2 expression compared to NT control cells ($p_{progenitor}$ = 0.064; $p_{mid-CM}$ = 0.0006; $p_{late\ CM}$ = 0.1, Mann-Whitney U) (*Figure 4N–O*).

In summary, we observe that perturbed cells express late-CM genes at lower levels and mid-CM genes at higher levels than NT control cells in the late-CM state. This suggests that although a subset of sgTBX5Enh cells can reach the late-CM cell state, these cells are in a delayed differentiation state that reflects mid-CM gene signatures. Taken together with previous observations, misregulation of

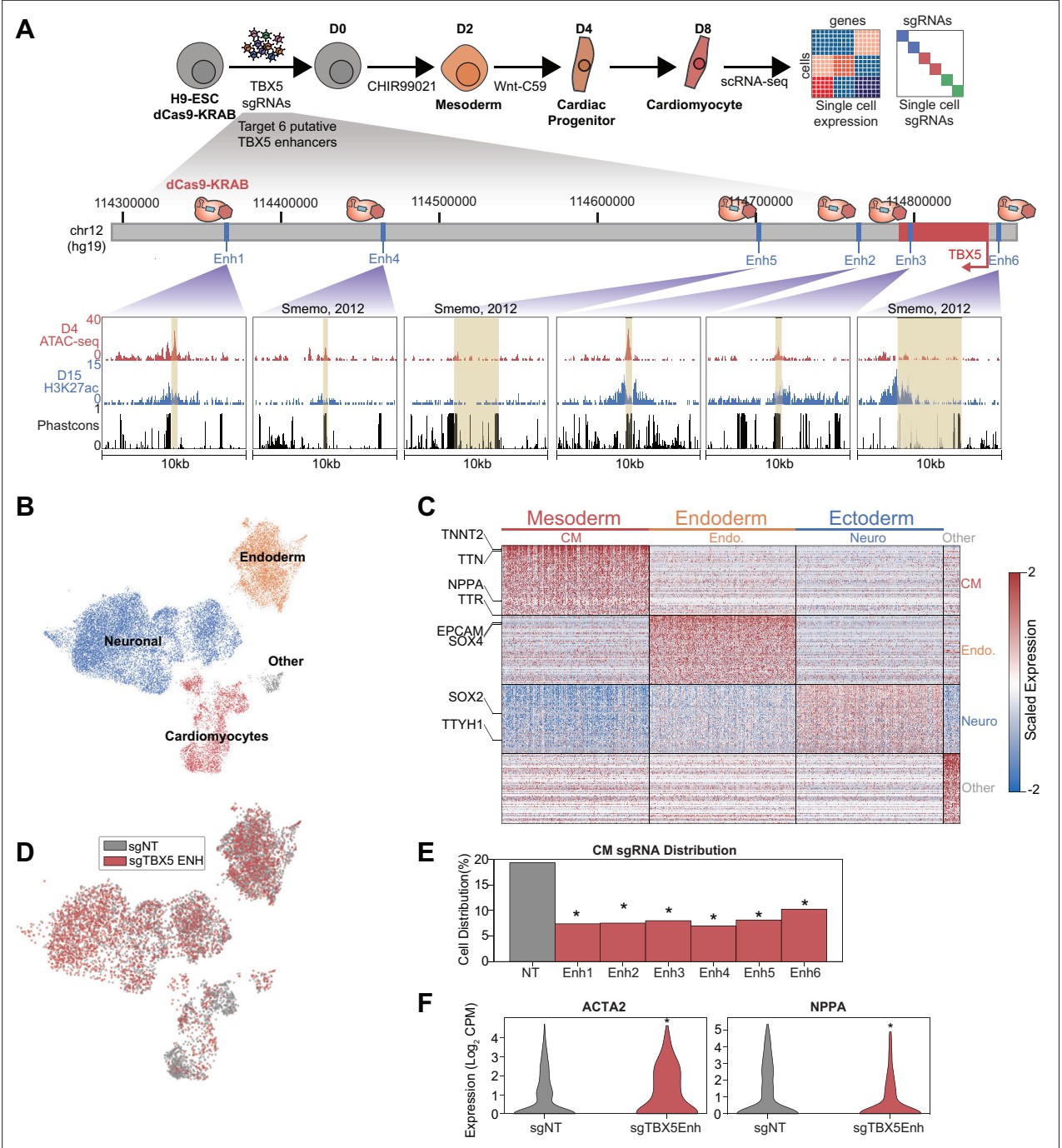

**Figure 3.** A focused validation screen demonstrates that TBX5 enhancers modulate cardiomyocyte (CM) cell fate. (**A**) (Top) Schematic of validation CRISPRi screen. H9-dCas9-KRAB cells are infected with lentiviral sgRNA library targeting TBX5 enhancers and differentiated over 8 days into CMs followed by scRNA-seq. (Bottom) Genome browser tracks of chromatin status and sequence conservation for TBX5 enhancers. Yellow regions denote enhancers. (**B**) UMAP visualization of H9-derived cells after 8 days of differentiation. Four Louvain clusters indicated. (**C**) Expression of the top 100 cluster-defining genes for each Louvain cluster. (**D**) Feature plots of cells receiving sgRNAs targeting TBX5 enhancers (red) or non-targeting (NT) control (gray). (**E**) Distribution of cells receiving sgNT and sgTBX5 enhancers that differentiate into CMs (*p<0.05 by hypergeometric test). (**F**) Expression of ACTA2 (left) and NPPA (right) in sgTBX5 enhancer and control sgNT CMs (*p<0.05 by Mann-Whitney U).

The online version of this article includes the following figure supplement(s) for figure 3:

**Figure supplement 1.** Focused screen marker expression and sgRNA distribution.

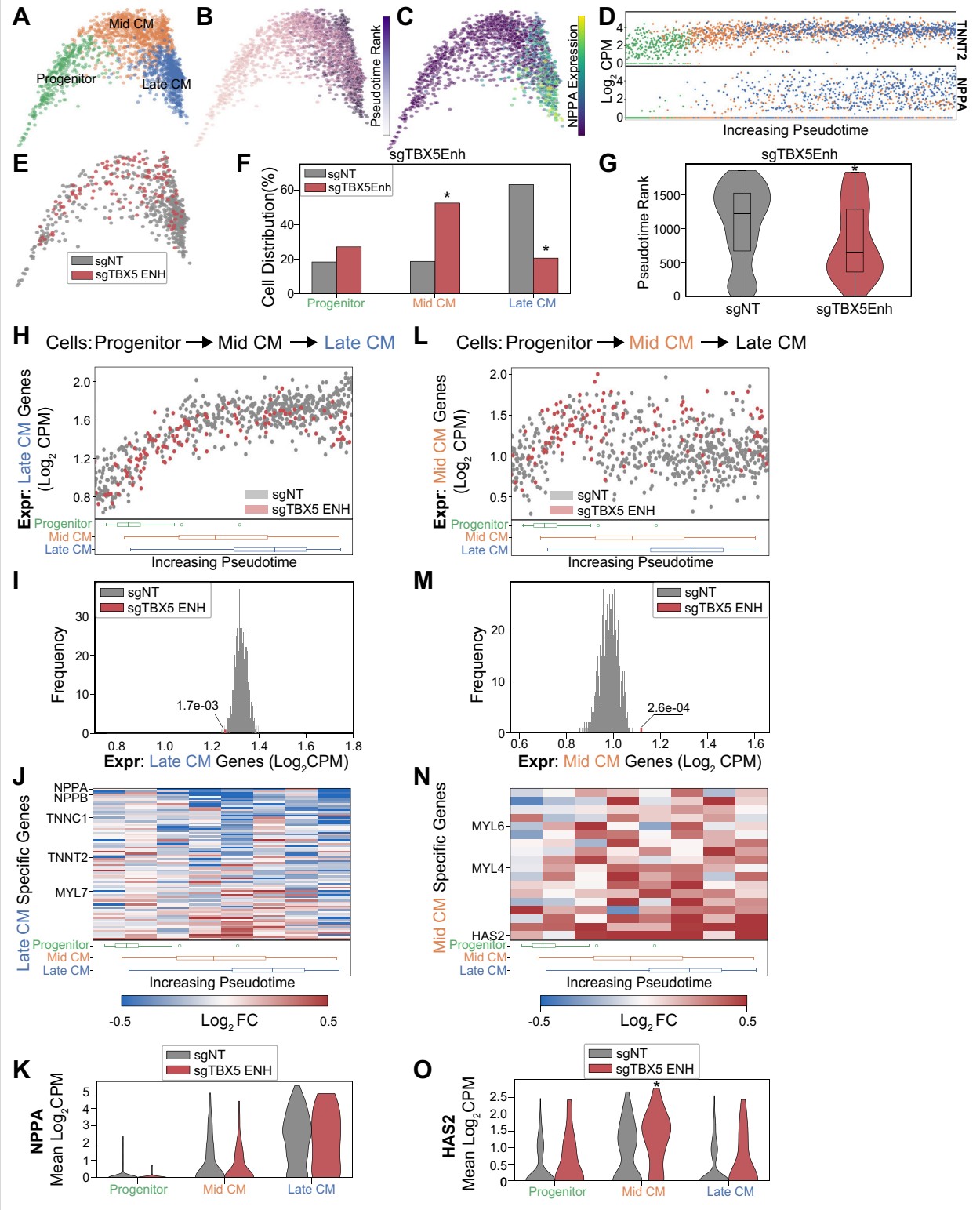

**Figure 4.** TBX5 enhancer repression alters cardiomyocyte (CM) molecular signatures. PHATE visualization of CMs in the focused screen with three Louvain clusters. (**B**) Feature plot of pseudotime ranking of cells across PHATE trajectory. (**C**) Feature plot of NPPA expression. (**D**) Single-cell expression of TNNT2 (top) and NPPA (bottom) across pseudotime ordered CMs. (**E**) Feature plots of cells receiving sgRNAs targeting TBX5 enhancers or non-targeting non-genome targeting (NT) control. (**F**) Distribution of states for cells receiving sgRNAs targeting TBX5 enhancers or non-targeting NT control (*p<0.05 by hypergeometric test). (**G**) Distribution of pseudotime rank for cells receiving sgRNAs targeting TBX5 enhancers and NCs (*p<0.05 by Mann-Whitney U) ($n_{NT}$ = 558 cells, $n_{TBX5\ Enh}$ = 122 cells). (**H**) (Top) We defined late-CM genes as those more expressed in late-CM cells. Shown is the

*Figure 4 continued on next page*

Figure 4 continued

average expression of late-CM genes in cells across pseudotime. Cells receiving sgRNAs targeting TBX5 enhancers (red) or non-targeting NT control (gray). (Bottom) Boxplots of CM subpopulation pseudotime ranks across pseudotime. (I) Average expression of late-CM genes across cells receiving sgRNAs targeting TBX5 enhancers (red) and 1000 random samplings of non-targeting control (sgNT) late-CM cells (gray) (*p<0.05 by Z-test). (J) (Top) For late-CM genes, shown is the relative expression in cells receiving sgRNAs targeting TBX5 enhancers compared with non-targeting NT control, across pseudotime. (Bottom) Boxplots of CM subpopulation pseudotime ranks across pseudotime. (K) NPPA expression in cells receiving sgRNAs targeting TBX5 enhancers and NC. (L) (Top) We defined mid-CM genes as those more expressed in mid-CM cells. Shown is the average expression of mid-CM genes in cells across pseudotime. Cells receiving sgRNAs targeting TBX5 enhancers (red) or non-targeting NT control (gray). (Bottom) Boxplots of CM subpopulation pseudotime ranks across pseudotime. (M) Averaged expression of mid-CM genes across cells receiving sgRNAs targeting TBX5 enhancers (red) and 1000 random samplings of non-targeting control (sgNT) late-CM cells (gray) (*p<0.05 by Z-test). (N) (Top) For mid-CM genes, shown is the relative expression in cells receiving sgRNAs targeting TBX5 enhancers compared with non-targeting NT control, across pseudotime. (Bottom) Boxplots of CM subpopulation pseudotime ranks across pseudotime. (O) HAS2 expression in cells receiving sgRNAs targeting TBX5 enhancers and non-targeting NT control.

The online version of this article includes the following figure supplement(s) for figure 4:

**Figure supplement 1.** Focused screen cardiomyocyte (CM) subpopulation marker expression and sgRNA distribution.

TBX5 through enhancer modulation leads to deficient induction of CM transcriptional signatures that are associated with CM fate commitment.

## TBX5 enhancer knockouts recapitulate cardiac phenotypes

Next, to confirm the results from the CRISPRi experiments above, we used a CRISPR/Cas9 strategy to genetically delete enhancers. We focused on the two TBX5 enhancers (Enh3 and Enh5) that harbor CHD-associated variants and exhibit the strongest cellular phenotypes in the CRISPRi screens. To compare enhancer deletion to gene deletion, we also knocked out TBX5 exon 3 (*Kathiriya et al., 2021*). We screened, isolated, and confirmed clones with biallelic deletion of either Enh3 (TBX5Enh3$^{-/-}$), Enh5 (TBX5Enh5$^{-/-}$), or Exon 3 (TBX5Exon$^{-/-}$)(*Figure 5A–B*). Consistent with the CRISPRi analysis, we find that all three knockouts can differentiate to CMs (*Figure 5—figure supplement 1A*). We observe that deletion of TBX5 enhancers phenocopies TBX5 gene deletion, with all knockout clones exhibiting reduced RNA expression by qPCR and protein expression by immunocytochemistry (*Figure 5C–E*).

Next, to test if genetic deletion of enhancers contributes to deficient CM specification, we repeated our scRNA-seq analysis at day 8 of CM differentiation (*Figure 5—figure supplement 1A* and *Supplementary file 3*). Mirroring the CRISPRi results, analysis of all cell populations shows a strong depletion of CM states in perturbed cells (*Figure 5F* and *Figure 5—figure supplement 1B*). TBX5Enh5$^{-/-}$ cells exhibited the strongest depletion (~9% CMs, p>2.2e-308, hypergeometric test) relative to WT control cells (~57% CMs), followed by TBX5Enh3$^{-/-}$ cells (~24% CMs, p=5.2e-263, hypergeometric test), and TBX5Exon$^{-/-}$ cells (~33% CMs, p=5.9e–166, hypergeometric test). Importantly, we observed consistent results over multiple biological replicates.

To examine if enhancer deletion also contributes to deficient CM differentiation, we focused our analysis on a cluster of TNNT2+ cells. Sub-clustering revealed three CM states: FN1+ early CMs, ACTA2+ mid-CMs, and NPPA+ late CMs (*Figure 5G*). While cells derived from each genetic background were represented across all three states (*Figure 5H* and *Figure 5—figure supplement 1C*), we observed notable differences. Notably, perturbed cells were on average 10.6-fold less likely to reach the mature late-CM state (~27% WT$_{Late\_CM}$,~5% Enh3$^{-/-}$$_{Late\_CM}$,~11% Enh5$^{-/-}$$_{Late\_CM}$,~1% Exon$^{-/-}$$_{Late\_CM}$) and this observation was statistically significant (p$_{Enh3}$=1.8e-24, p$_{Enh5}$=1.6e-7, p$_{Exon}$ = 5.8e-83) (*Figure 5I*). Next, we repeated our previous analysis to confirm that cells with TBX5 enhancer or exon deletion exhibit elevated expression of mid-CM genes (*Figure 5J*). Importantly, deletion of either TBX5 enhancer or exon 3 resulted in the repression of key genes of cardiac development including the TBX5 target gene NPPA (*Figure 5K*, *Figure 5—figure supplement 1D*, and *Supplementary file 4*; *Houweling et al., 2005*). To confirm this result using an orthogonal assay, we performed RNA fluorescence in situ hybridization coupled to a flow cytometry readout (FlowFISH) (*Fulco et al., 2019*; *Reilly et al., 2021*). Consistent with an increase in earlier stage CMs, we find that TBX5 gene and enhancer knockouts have increased RNA and protein expression of early CM markers such as FN1 (*Figure 5L–N* and *Figure 5—figure supplement 1E*). Also, consistent with depletion in later stage CMs, we observe that TBX5 gene and enhancer deletions have decreased RNA and protein expression of the late-CM marker NPPA (*Figure 5C–D*).

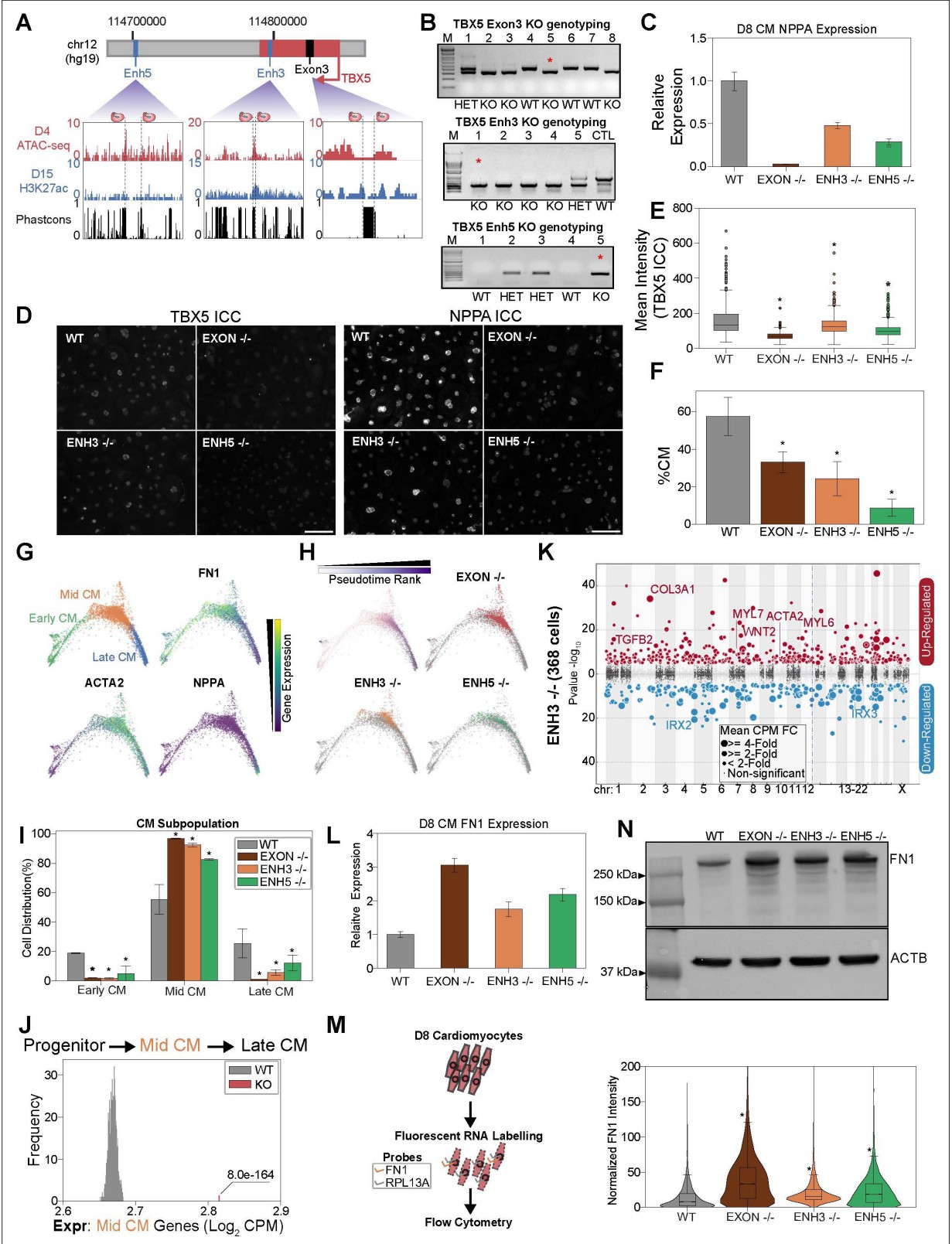

**Figure 5.** TBX5 enhancer knockouts recapitulate CRISPRi phenotypes. (**A**) (Top) TBX5 enhancer knockout strategy. (Bottom) Genome browser snapshots of chromatin status and sequence conservation. Dotted lines denote sgRNA target sites. (**B**) Genotyping PCR to verify TBX5 knockouts of exon 3 (top), enhancer 3 (middle), and enhancer 5 (bottom). Red asterisk indicates clones used in downstream analysis. (**C**) NPPA transcript expression in exon and enhancer knockout cells after 8 days of cardiomyocyte (CM) differentiation. qPCR quantification normalized to reference gene (RPLP0) and

Figure 5 continued

then compared with WT cells (n = 3 technical replicates and 2 biological replicates). (**D**) ICC for TBX5 (left) and NPPA (right) in WT and TBX5 exon 3, enhancer 3, and enhancer 5 knockout cells (Scale bar = 100μm). (**E**) Quantification of TBX5 ICC (mean intensity) across TBX5 genotyping (*p<0.05 by Mann-Whitney U) ($n_{WT}$ = 2003 cells, $n_{EXON\ -/-}$ = 780 cells, $n_{ENH3\ -/-}$ = 1357 cells, $n_{ENH5\ -/-}$ = 1592 cells). (**F**) Distribution of WT, TBX5 enhancer KO, or exon KO cells that differentiate into CMs relative to endoderm population (*p<0.05 by hypergeometric test) ($n_{WT}$ = 2 biological replicates, $n_{EXON\ -/-}$ = 2 biological replicates, $n_{ENH3\ -/-}$ = 2 biological replicates, $n_{ENH5\ -/-}$ = 2 biological replicates). (**G**) (Top left) PHATE visualization of CMs with three Louvain clusters. (Other quadrants) Feature plots of FN1, ACTA2, and NPPA expression. (**H**) (Top left) Feature plot of pseudotime ranking of CM cells across PHATE trajectory. (Other quadrants) Distribution of TBX5 exon KO and enhancer KO cells across CM trajectory. WT: gray. (**I**) Distribution of WT, TBX5 exon KO, and enhancer KO cells across three CM subpopulations (*p<0.05 by hypergeometric test) ($n_{WT}$ = 2 biological replicates, $n_{EXON\ -/-}$ = 2 biological replicates, $n_{ENH3\ -/-}$ = 2 biological replicates, $n_{ENH5\ -/-}$ = 2 biological replicates). (**J**) Averaged expression of mid-CM genes across TBX5 exon and enhancer KO cells (red) and 1000 random samplings of WT mid-CM cells (gray) (*p<0.05 by Z-test). (**K**) Differentially expressed genes in enhancer 3 KO cells in CM states. Please see description of Manhattan plot in **Figure 2I**. (**L**) FN1 transcript expression in exon and enhancer knockout cells after 8 days of CM differentiation. qPCR quantification normalized to reference gene (RPLP0) and then compared with WT cells (n = 3 technical replicates). (**M**) (Left) Overview of FlowFISH experiment. (Right) Flow cytometry of FN1 RNA FISH intensity normalized by control RPL13A intensity in TBX5 WT and KO lines (*p<0.05 by Mann-Whitney U) ($n_{WT}$ = 4312 cells, $n_{EXON\ -/-}$ = 4034 cells, $n_{ENH3\ -/-}$ = 11166 cells, $n_{ENH5\ -/-}$ = 12286 cells). (**N**) (Top) FN1 and ACTB (bottom) protein expression in WT, TBX5 exon, and enhancer KO cells after 8 days of CM differentiation.

The online version of this article includes the following source data and figure supplement(s) for figure 5:

**Source data 1.** Original genotyping gels for panel B.

**Source data 2.** Original genotyping gels for panel B.

**Source data 3.** Original genotyping gels for panel B.

**Source data 4.** Original genotyping gels for panel B.

**Source data 5.** Original genotyping gels for panel B.

**Source data 6.** Original genotyping gels for panel B.

**Source data 7.** Original western blots for panel N. Western blot of actin control, with labels.

**Source data 8.** Original western blots for panel N. Western blot of actin control, without labels.

**Source data 9.** Original western blots for panel N. Western blot of FN1, with labels.

**Source data 10.** Original western blots for panel N. Western blot of FN1, without labels.

**Figure supplement 1.** TBX5 enhancer knockout cell distribution.

Finally, since our analyses also generated clones with heterozygous TBX5 enhancer deletion (**Figure 5B**), we performed further analysis of one clone: TBX5Enh5$^{+/-}$ (**Figure 6A–B**). Confirming that these genetic modifications alter TBX5 expression, we observed that clones with enhancer deletions exhibit significantly decreased RNA expression during CM differentiation (**Figure 6C**). After CM differentiation and scRNA-seq (**Figure 6D** and **Figure 6—figure supplement 1A**), TBX5 Enh5 heterozygous knockouts also exhibited significant depletion of CM states (~24% CMs, p=5.2e-125, hypergeometric test) relative to WT cells (~45% CMs) (**Figure 6E** and **Figure 6—figure supplement 1B**). Sub-clustering of a TNNT2+ cluster revealed four CM states: FN1+ early CMs, ACTA2+ mid-CMs, HAND1+ late CMs, and IRX2+ ventricular CMs (**Figure 6F** and **Figure 6—figure supplement 1C**). Consistently, focused pseudotime analysis of CMs also showed significant depletion of Enh5$^{+/-}$ cells in the mature ventricular CM state (~39% WT$_{Ven\_CM}$, ~6% Enh5$^{+/-}$$_{Ven\_CM}$, p=1.5e-75) (**Figure 6G–H**). Importantly, heterozygous deletion of TBX5 Enh5 results in a weaker differentiation defect than homozygous deletion of TBX5 Enh3 (which has a weaker phenotype compared to homozygous deletion of TBX5 Enh5) (**Figure 6G–H**). These observations are consistent with those observed for the TBX5 gene deletions previously published (**Kathiriya et al., 2021**). Next, we repeated our previous analysis to confirm that cells with TBX5 enhancer knockout exhibit delayed activation of late-CM genes (**Figure 6I**) and delayed repression of mid-CM genes (**Figure 6J**). Our analysis also revealed a depletion of cardiac related genes in perturbed late-CM cells compared to controls (**Figure 6K** and **Supplementary file 6**). Cell label transfer onto all previous datasets showed expected clustering consistent deficiencies to achieve later CM states (**Figure 6—figure supplement 1D–G**).

In sum, these observations with TBX5 enhancer deletion cells are consistent with our CRISPRi studies, and support the conclusion that loss of TBX5 enhancer function results in deficient transcriptional specification of CMs.

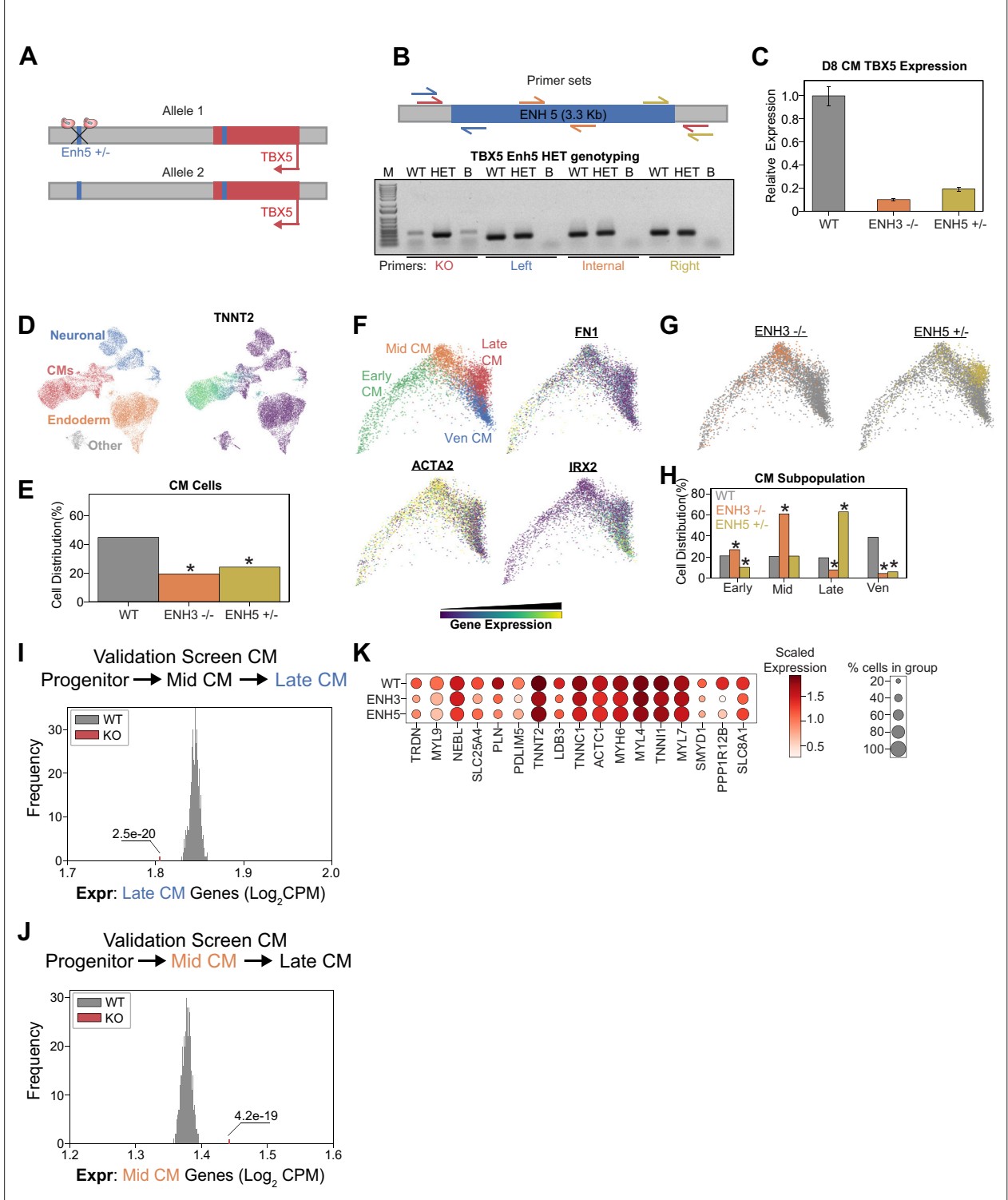

**Figure 6.** Heterozygous TBX5 enhancer 5 knockout (KO) displays reduced phenotypes. (**A**) TBX5 enhancer 5 heterozygous KO strategy. (**B**) (Top) Position of primers used to validate TBX5 enhancer 5 heterozygous clone. Red: KO enhancer-spanning primers; blue: left junction primers; orange: enhancer internal primers; and yellow: right junction primers. (Bottom) Genotyping PCR to verify TBX5 enhancer 5 heterozygous deletion. Like the WT, the heterozygous clone retains left, internal, and right fragments, consistent with retaining a copy of enhancer 5. Unlike the WT, the heterozygous clone yields a small fragment when amplified with the KO enhancer-spanning primers. PCR conditions were not optimized for amplification of the WT large 3-kb+ fragment. WT: wild-type; HET: TBX5 enhancer 5 heterozygous clone; B: blank. (**C**) TBX5 transcript expression in enhancer KO cells after 8 days of cardiomyocyte (CM) differentiation. qPCR quantification normalized to reference gene (RPLP0) and then compared with control cells (n = 3 technical

*Figure 6 continued on next page*

*Figure 6 continued*

replicates). (**D**) (Left) UMAP visualization of H9-derived cells after 8 days of differentiation. Four Louvain clusters indicated. (Right) Feature plot of TNNT2 expression. (**E**) Distribution of WT and TBX5 enhancer KO cells after differentiation (*$p<0.05$ by hypergeometric test). (**F**) (Top left) PHATE visualization of CM trajectory cells with four distinct Louvain clusters. (Other quadrants) Feature plot of FN1, ACTA2, and IRX2 expression. (**G**) Distribution of TBX5 enhancer KO cells across CM trajectory. WT: gray. (**H**) Cell distribution of TBX5 enhancer KO cells across four CM subpopulations (*$p<0.05$ by hypergeometric test). (**I**) Averaged expression of late-CM genes across TBX5 enhancer KO cells (red) and 1000 random samplings of WT late-CM cells (gray) (*$p<0.05$ by Z-test). (**J**) Averaged expression of mid-CM genes across TBX5 enhancer KO cells (red) and 1000 random samplings of WT late-CM cells (gray) (*$p<0.05$ by Z-test). (**K**) Dotplot shows the expression of cardiac genes in WT and KO cells belonging to the late-CM cluster.

The online version of this article includes the following source data and figure supplement(s) for figure 6:

**Source data 1.** Original genotyping gels for panel B. Enhancer 5 HET genotyping, with labels.

**Source data 2.** Original genotyping gels for panel B. Enhancer 5 HET genotyping, without labels.

**Figure supplement 1.** TBX5 heterozygous enhancer knockout validation.

## Discussion

This study identifies and functionally characterizes 16 enhancers that are required for normal CM differentiation. Perturbations of these enhancers yield diverse cellular phenotypes ranging from global reduction of mesoderm populations to deficient CM differentiation. These cell specification defects are consistent with the putative role of these enhancers in regulating known cardiac regulators such as TBX5 (*Kathiriya et al., 2021*). The diversity of cellular responses has several implications. First, our observations highlight the important role of enhancers in orchestrating spatiotemporal gene regulation in development (*Plank and Dean, 2014*). Second, that enhancer perturbation can lead to such diverse cellular consequences further highlights the potential role of genetic variants in modifying enhancer function and contributing to developmental defects such as CHD (*Richter et al., 2020*; *Smemo et al., 2012*). We speculate that focused sequencing of genetic variants at these enhancers will reveal new genetic contributors of CHD. Third, the diversity of cellular responses observed through enhancer perturbation suggests that the cellular phenotypes of CHD could also be very diverse, and that subtle changes in cell fate may lead to complex CHD phenotypes. Indeed, CHD is a diverse developmental disease with multiple causal genes and distinct disease subtypes (*van der Linde et al., 2011*).

TBX5 presents a potential framework for studies to understand the genetic basis of human developmental disorders by linking human genetic linkage studies (strongest evidence of variant function) to variants of unknown significance (least support). For example, coding variants at TBX5 cause cardiac defects such as Holt-Oram syndrome (*Basson et al., 1997*; *Bruneau et al., 1999*; *Holt and Oram, 1960*; *Li et al., 1997*). Consistently, gene knockout studies in human iPSCs have shown that dosage-sensitive impairment of TBX5 alters CM differentiation with CHD-relevant phenotypes (*Kathiriya et al., 2021*). Even though multiple enhancers control TBX5 expression (*Smemo et al., 2012*), our studies show that perturbation of TBX5 enhancers exhibit similar cell state phenotypes. Enhancers in close proximity to genes with known roles in CHD represent key targets for future investigation.

Interestingly, our results indicate that knockdown of TNNT2 results in transcriptional phenotypes. While TNNT2 is not known to function as a transcriptional regulator, recent results from similar screens have identified unexpected gene expression phenotypes for genes not typically associated with regulatory function. For example, *Replogle et al., 2022* showed that knockdown of chromosome segregation genes (with unclear roles in transcription) resulted in gene expression phenotypes. Similarly, genes with roles in glycolysis, vesicular trafficking, and DNA replication also unexpectedly caused transcriptional changes. These observations make possible the systematic assessment of genotype-phenotype relationships by using the transcriptome as a readout. Thus, TNNT2 knockdown may impact transcriptional states through similarly unexpected effects. One possibility is that, since TNNT2 is a critical component of the sarcomeric apparatus, its loss could initiate a feedback response resulting in delayed CM specification. In support of this possibility, we observe that CRISPRi perturbations of two TNNT2 enhancers and the promoter consistently yield changes in transcriptional and cell state phenotypes (*Figure 2G and I*).

This study represents a proof of concept for future efforts to systematically test the function of enhancers in heterogeneous developmental systems. We adopted a tiered approach to balance the advantages and disadvantages of CRISPR and CRISPRi approaches (*Diao et al., 2017*; *Fulco et al.,*

*2016*; *Gasperini et al., 2017*; *Thakore et al., 2015*; *Zhou et al., 2014*). That is, while CRISPRi-based epigenetic silencing can efficiently act across many cells in a population, it offers incomplete inhibition. In contrast, the generation of genetic knockouts using CRISPR is relatively inefficient, but yields clones with complete enhancer deletion. Thus, we started the study with a higher throughput single-cell CRISPRi screens as a filter for potential hits. To ensure robustness, we validated these hits with a smaller scale CRISPRi validation screen. Finally, to ensure consistency with orthogonal perturbation paradigms, we performed clonal enhancer deletion studies. This tiered approach allowed us to balance speed with accuracy. Tiering strategies such as this and others (*Klann et al., 2021*) will aid future efforts. Importantly, we found that consistent cellular phenotypes can be obtained with both CRISPRi or CRISPR strategies. Expectedly, perturbations of promoters generally yield stronger cellular responses than enhancers. Similarly, genetic knockouts elicit more dramatic changes in cell state than CRISPRi with dCas9-KRAB. Genetic variants likely yield even weaker phenotypes. In this way, we speculate that epigenetic perturbation more closely matches variants than genetic deletion.

Despite the overall success of our screening approach and subsequent validation experiments, we note several potential limitations. First, scRNA-seq provides deep cellular coverage at the expense of sequencing depth per cell, such that detection of lowly expressed genes is challenging. Thus, conclusive determination of sgRNA functionality in individual cells remains problematic. Second, enhancers are known to function in a time-dependent manner, yet sgRNA perturbations were introduced at a fixed time point. While we expect time-dependent perturbations to yield insights on enhancer-mediated gene regulation, one technical challenge is that the temporal delay between perturbation induction and functional repression needs to be resolved and likely improved before application in dynamic developmental systems. Third, CM differentiation in two-dimensional culture systems does not adequately recapitulate the complex morphological and inter-cell communication events that occur during human development. Standard cardiac differentiation strategies are very efficient but use harsh metabolic selection to enrich for CMs. In our study, we intentionally avoided metabolic selection to preserve non-cell autonomous interactions, since selection may inadvertently mask the phenotypes of genetic perturbations. By eliminating the metabolic selection step during hESC differentiation, we produced heterogeneous cell states including mesodermal (CMs and progenitors), ectodermal (neuronal), and endodermal states. This strategy was previously used by *Tohyama et al., 2013*. In support of the relevance of this system, Zic2 is required for early specification of CMs. In our system, CRISPRi knockdown of Zic2 results in loss of CM and neuronal specification. These cells with Zic2 sgRNAs are almost entirely diverted to an early mesodermal state (*Figure 1—figure supplement 1H*). These results indicate that our biological system captures expected phenotypes and suggest that the results from other perturbations are also potentially relevant to CHD. Future utilization of organoid systems is likely to better mimic the complex morphogenetic changes that occur in vivo.

In summary, we have applied single-cell screening technology to functionally characterize enhancers in a heterogeneous developmental system. We expect that future applications of these approaches will help to comprehensively identify enhancers with roles in cell fate specification that could also contribute to developmental defects such as CHD.

## Materials and methods

**Key resources table**

| Reagent type (species) or resource | Designation | Source or reference | Identifiers | Additional information |
|---|---|---|---|---|
| Chemical compound, drug | Puromycin | Cayman Chemical | Cat#13884 | |
| Chemical compound, drug | Blasticidin | RPI | Cat#3513-03-9 | |
| Chemical compound, drug | Thiazovivin | Sigma-Aldrich | Cat#SML1045 | |
| Chemical compound, drug | TrypLE Select | Thermo Fisher | Cat#12563 | |

*Continued on next page*

*Continued*

| Reagent type (species) or resource | Designation | Source or reference | Identifiers | Additional information |
|---|---|---|---|---|
| Chemical compound, drug | CHIR99021 | Tocris | Cat#4423 | |
| Chemical compound, drug | Wnt-C59 | Cayman | Cat#16644 | |
| Chemical compound, drug | Accutase | Sigma-Aldrich | Cat#SCR005 | |
| Chemical compound, drug | Insulin | Gibco | Cat#12585014 | |
| Chemical compound, drug | B-27 | Thermo Fisher | Cat#17504044 | |
| Chemical compound, drug | Molecular Probes Fura-2 | Thermo Fisher | Cat#F1221 | |
| Chemical compound, drug | Pluronic F-127 | Thermo Fisher | Cat#P6867 | |
| Antibody | Anti-human-FN1 (Rabbit polyclonal) | Thermo Fisher | Cat#PA5-29578 | WB(1:1000) |
| Antibody | Anti-rabbit IgG | Cell Signaling Technology | CST #7074 | WB(1:400) |
| Chemical compound, drug | KAPA HiFi HS | KAPA | Cat#KK2502 | |
| Commercial assay, kit | PrimeFlow RNA Assay Kit | Thermo Fisher | Cat#88-18005-210 | |
| Chemical compound, drug | NEBNext High-Fidelity | New England Biolabs | Cat#M0541L | |
| Chemical compound, drug | Gibson Assembly Master Mix | New England Biolabs | Cat#E2611L | |
| Chemical compound, drug | mTeSR Plus | Stemcell Technologies | Cat#100–0276 | |
| Chemical compound, drug | Matrigel | Corning | Cat#354277 | |
| Commercial assay, kit | P3 Primary Cell 4D-Nucleofector X Kit | Lonza | Cat#V4XP-3024 | |
| Commercial assay, kit | 10× genomics Chromium Single Cell 3' Kit V3.1 | 10× Genomics | Cat#PN-1000147 | |
| Commercial assay, kit | 10× CellPlex | 10× Genomics | Cat#PN-1000261 | |
| Commercial assay, kit | 10× Target Hybridization Kit | 10× Genomics | Cat#PN-1000248 | |
| Chemical compound, drug | RPMI 1640 | Thermo Fisher | Cat#11875093 | |
| Chemical compound, drug | KnockOut Serum | Thermo Fisher | Cat#10828028 | |
| Chemical compound, drug | HHBSS | Corning | Cat#21-023-CM | |
| Strain, strain background (Endura) | Endura ElectroCompetent Cells | Lucigen | Cat#60242–2 | Electrocompetent Cells |
| Strain, strain background (*Escherichia coli*) | Stellar Competent Cells | Clontech | Cat#636766 | |
| Other | Single-cell RNA-seq Data | This paper | GEO: GSE190475 | Sequencing data located at GEO |

*Continued on next page*

*Continued*

| Reagent type (species) or resource | Designation | Source or reference | Identifiers | Additional information |
|---|---|---|---|---|
| Cell line (*Homo sapiens*) | 293T cells | ATCC | CRL-3216 | Mycoplasma free; ATCC STR authenticated as CRL-3216 |
| Cell line (*Homo sapiens*) | H9 cells | WiCell | WA09 | Mycoplasma free; ATCC STR authenticated as WAe009-A-18 |
| Recombinant DNA reagent | Plasmid: pMD2.G | Addgene | RRID:Addgene_12259 | |
| Recombinant DNA reagent | Plasmid: psPAX2 | Addgene | RRID:Addgene_12260 | |
| Recombinant DNA reagent | Plasmid: lenti-dCas9-KRAB-Blast | Addgene | RRID:Addgene_89567 | |
| Recombinant DNA reagent | Plasmid: CROPseq-Guide-puro | Addgene | RRID:Addgene_86708 | |
| Sequence-based reagent | sgRNA oligos | This paper | sgRNA oligos | *Supplementary file 1* |
| Sequence-based reagent | qPCR primers | This paper | qPCR primers | *Supplementary file 2* |
| Software, algorithm | Star | PMID:23104886 | RRID:SCR_004463 | |
| Software, algorithm | Picard | Broad Institute | RRID:SCR_006525 | |
| Software, algorithm | FlowCal | PMID:27110723 | RRID:SCR_018140 | |
| Software, algorithm | FeatureCounts | DOI:10.1093/bioinformatics/btt656 | RRID:SCR_012919 | |
| Software, algorithm | 10× Genomics Cellranger | 10× Genomics | RRID:SCR_023221 | |
| Software, algorithm | Scanpy | PMID:29409532 | RRID:SCR_018139 | |
| Software, algorithm | IGV | Broad Institute | RRID:SCR_011793 | |
| Other | Illumina NextSeq 500 instrument | Illumina | | Next-generation sequencer |
| Other | Illumina NextSeq 2000 instrument | Illumina | | Next-generation sequencer |
| Other | Illumina NovaSeq 6000 instrument | Illumina | | Next-generation sequencer |
| Other | Agilent 2200 TapeStation instrument | Agilent | | Automated electrophoresis instrument |
| Other | Qubit Fluorometric Quantitation instrument | Thermo Fisher | | Flurometer |
| Other | EVOS FL Auto Imaging System | Thermo Fisher | | Microscope |

## Materials availability statement

Data and codes required to reproduce the findings of this manuscript are freely available and are referenced below. Further information and requests for resources and reagents should be directed to and will be fulfilled by Gary Hon (Gary.Hon@utsouthwestern.edu).

## Generation of PiggyBac dCas9-KRAB plasmid

The NheI/SalI dCas9-KRAB fragment from Lenti-dCas9-KRAB-blast (Addgene ID: 89567) was ligated into PiggyBac vector 5'-PTK-3' (*Cadiñanos and Bradley, 2007*) replacing the NheI/SalI insert. A bovine growth hormone polyadenylation signal was PCR amplified (Kapa HiFi DNA polymerase) (5'-GCC TCC CCG CAT CGA TAC CGC TGT GCC TTC TAG TTG CCA G, 3'-GTA ACA AAA CTT TTA ACT AGC CA T AGA GCC CAC CGC ATC C) and assembled (NEB HiFi Assembly) replacing the remaining SalI/SpeI fragment. A 647 bp ubiquitous chromatin opening element derived from the human HNRPA2B1-CBX3

locus was PCR amplified (5'-TAT AGA TAT CAA CTA GAA TGG GGA GGT GGT CCC TGC AG, 3'-A AC TTT ATC CAT CTT TGC AGG GCC CTC CGC GCC TAC AG) and assembled into the NheI site.

## Target selection

Fourteen enhancers were selected using publicly available ATAC-seq and H3K27ac ChIP-seq datasets collected at various time points throughout CM differentiation. ATAC-seq peaks across all time points were merged and extended to a total size of 500 bp. We then applied FeatureCounts for H3K27ac ChIP-seq signal at these 500 bp ATAC-seq boundaries. H3K27ac counts were normalized to RPKM then to input followed by $\log_2$ transformation. We selected for ATAC-seq regions in which the max ChIP enrichment across all time points was greater than $\log_2(1.5)$, identifying 43,536 enhancers across CM differentiation. We then selected enhancers which were within 1 kb of a CHD-associated variant which left 2300 CHD-associated enhancers. Using gene ontology, we identified enhancers which were within 100 kb of a heart development-associated gene. After manually filtering for genes based on prior literature, we were left with 14 enhancers. We also included the 18 nearby promoters for heart development genes. Our library also consisted of six TBX5 enhancers, three we identified in house using the same criteria above with the expectation of CHD variant overlap, and three from literature (*Smemo et al., 2012*). From an MPRA study, we also selected five enhancers in which CHD-associated variants have been shown to perturb activity (*Richter et al., 2020*).

## Generation of sgRNA library

We used CROPseq-Guide-Puro plasmid for sgRNA expression (Addgene ID: 86708). For large-screen sgRNA library construction, a single-strand sgRNA oligo library containing 397 sgRNAs (*Supplementary file 1*) was synthesized by IDT. The library was amplified by NEBNext High-Fidelity 2× PCR master mix (New England Biolabs) to make it double-stranded and then was inserted into the BsmBI digested CROPseq-Guide-Puro plasmid through Gibson Assembly (New England Biolabs). The circulized product was purified and electroporated into Endura ElectroCompetent cells (Lucigen) following the manufacturer's protocol. The cells were then cultured in LB medium with 100 µg/ml ampicillin at 30°C overnight and the plasmid extracted using the ZymoPURE plasmid maxiprep kit (Zymo Research). We amplified the spacer sequences of the sgRNA library and verified the complexity of the library by Illumina sequencing. See our previous publication for a full protocol including primer sequences (*Xie et al., 2019*). Our focused screen consisted of 49 sgRNA which were packaged using a golden gate strategy previously published. In brief, single-strand sgRNA oligos were ordered from IDT and annealed using T4 ligase. Annealed oligos for sgRNA targeting the same region were then pooled for a golden gate reaction using BsmBI and T7 ligase to insert the sgRNA into the CROPseq backbone. Individual sgRNA were packaged in a similar approach.

## Cell line

H9 ESCs were maintained feeder-free in mTeSR Plus (Stemcell Technologies 100-0276) according to the suggested manufacturer's instructions on Matrigel (Corning 354277) coated plates, passaged every 3–4 days with DPBS/0.5 mM EDTA and plated in 2 µM thiazovivin (Sigma SML1045) supplemented mTeSR Plus overnight. H9 ESCs and HEK293Ts were authenticated with ATCC using STR (short tandem repeat) profiling. Both lines were verified as mycoplasma free through PCR detection (Bulldog Bio 25234).

## Generation of dCas9-KRAB-hESC line

H9 ESCs were washed with DPBS and dissociated to single cells with TrypLE Select. Transfection of $1 \times 10^6$ cells with 5 µg PiggyBac transposon plasmid and 1 µg HA-mPB (*Cadiñanos and Bradley, 2007*) transposase plasmid in 100 µl was performed using P3 Primary Cell 4D-Nucleofector X Kit (V4XP-3024) with program CB-150 according to the suggested manufacturer's instructions. At confluency cells were passaged and maintained in the presence of 5 µg/ml blasticidin S until emerging colonies were ~1 mm in diameter. Individual colonies were isolated, expanded, and tested for stable expression.

## CM differentiation

At 80–90% confluency, media was replaced with 3 µM CHIR99021 supplemented CDM3 (RPMI 1640; 0.5 mg/ml human albumin, ScienCell OsrHSA; 211 µg/ml L-ascorbic acid 2-phosphate) for 48 hr

followed by 2 µM Wnt-C59 supplemented CDM3 for 48 hr with subsequent media changes every 2 days with CDM3 alone. On day 8 post-differentiation, samples were washed with DPBS and dissociated to single cells with Accutase and resuspended in CDM3.

## Virus packaging

Lentiviral plasmid was packaged as previously described (*Xie et al., 2019*). Briefly, the lentivirus packaging plasmids PMD2.G and psPAX2 (Addgene ID: 12259 and 12260) were co-transfected with the carrier plasmid to HEK293T cells with linear polyethylenimine (Polysciences). Supernatant was collected 72 hr after transfection and filtered with a 0.45 µm filter. The virus was further purified by using Lenti-X lentivirus concentrator (Clontech).

## sgRNA transduction

Single-cell dissociated H9 ESCs ($1\times10^6$ cells) were incubated in 2 ml medium supplemented with thiazovivin and lentiviral supernatant (MOI ~ 0.3) in a single well of an ultra-low attachment plate (Corning 3471) for 3 hr, diluted in 2 ml medium and then plated onto three wells of a Matrigel-coated six-well plate. At 72 hr post-transduction cells were fed with 5 µg/ml blasticidin S and 1 µg/ml puromycin supplemented medium for 7 days.

## Knockout line generation

Transfection of 10 µg Cas9/sgRNA plasmids (left and right flanking at 1:1) was performed as described in stable line generation. At 48 hr post-transfection, single GFP-positive cells were sorted, expanded, and isolated for knockout validation. For validation, genomic DNA was extracted with QuickExtract (Lucigen) and PCR was performed to verify deleted regions.

## Immunocytochemistry

Cells were differentiated or passaged on Matrigel coated #1.5 12 mm cover glass. Cells were washed with DPBS, fixed with 4% PFA in DPBS for 20 min at room temperature, and washed with DPBS. Samples were blocked for 1 hr with PBS supplemented with 5% normal donkey serum and 0.3% Triton X-100. Primary antibodies in PBS supplemented with 1% BSA and 0.3% Triton X-100 were incubated overnight at 4°C. After 3×5 min washes with PBS, secondary antibodies at 1:400 were incubated at room temperature for 2 hr, washed, mounted with Prolong Glass (Invitrogen P36981), and imaged with an EVOS FL Auto Imaging System (Thermo Fisher).

## Western blotting

Whole-cell protein lysate was collected using RIPA buffer in the presence of protease inhibitors and PMSF. After protein concentration was determined, the lysate was denatured in 2× Laemmli Sample Buffer, separated by electrophoresis, and transferred to nitrocellulose membrane. Membranes were blocked with 5% non-fat dry milk in TRIS buffer containing 0.1% Tween-20 (TBST) for 1 hr followed by incubation with primary antibody overnight at 4°C. The membranes were then washed with TBST followed by application of appropriate HRP-conjugated secondary antibody for 1 hr at room temperature. After washing with TBST, membranes were incubated with Pierce SuperSignal West Femto and images with a Bio-Rad ChemiDoc Imager.

## FlowFISH

TBX5 exon 3, enhancer 3, enhancer 5, and WT H9 hESC were differentiated into CMs over 8 days followed by PrimeFlow following the commercial protocol. Each sample was multiplexed for FN1 (Alexa 647) and RPL13A (Alexa 488) probes followed by flow cytometry. Data was analyzed using the FlowCal python package (*Castillo-Hair et al., 2016*). Gating was performed using the density function along the forward scatter and side scatter axes. Briefly, the function takes in the percentage of events to retain as an input. Then, the plot is divided into grids and events in those grids are counted. A histogram is plotted, and the curve is smoothened using Gaussian blur. For this analysis, 45% of events were retained and further analysis was done using gated data. To normalize FN1, FN1 probe intensity was divided with RPL13A probe intensity at a single-cell level for all the samples.

## Single-cell RNA-seq

For the focus screen, dissociated CMs were resuspended in 0.08% BSA in PBS and passed through a 70 μm filter before diluting to 1000 cells/μL. Two 10× lanes were run at an expected recovery of 10,000 cells. For the initial screen, dissociated CMs were divided into 10 samples consisting of $5*10^5$ cells and each sample stained with 1 of 10 cell hashing bodies. Cells were processed as described in cell hashing publication (*Stoeckius et al., 2018*). After processing, cells were resuspended in 0.04% BSA in PBS and diluted to 4800 cells/μL. Eight 10× lanes were run at expected recovery of 45,000 cells. For sequencing exp (*Figure 5* and *Figure 6*), divided knockout and control dissociated CMs into 12 samples of $1*10^6$ cells and stained each using a unique CellPlex oligo. Samples were processed as in 10× Genomics Chromium 3' V3.1 with Cell Multiplexing kit and pooled together evenly for a final concentration of 1500 cells/μL in 0.04% BSA in PBS. For sequencing exp (*Figure 5*), two 10× lanes were run at an expected recovery of 30,000 cells. Single-cell RNA-seq libraries were prepared using 10× Genomics Chromium 3' V3.1 kit, following the standard protocol. For sequencing exp (*Figure 6*), two 10× lanes were run at an expected recovery of 60,000 cells. Single-cell RNA-seq libraries were prepared using 10× Genomics Chromium 3'V3.1 HT kit, following the standard protocol. To construct the sgRNA enrichment libraries, 50 ng of cDNA product was used to perform an enrichment PCR using SI-PCR primer and sgRNA enrichment primers. The PCR product was purified using a 1.6× SPRI beads cleanup followed by a final index PCR using SI-PCR primer and Nextera primers. A final 1.6× SPRI beads cleanup was then performed with a final expected library size of 500 bp. To construct cell hashing libraries, 20 μL of supernatant fraction cDNA was used for hashtag amplification PCR using SI-PCR primer and Nextera Hashtag primers. For a more detailed protocol, see cell hashing publication (*Stoeckius et al., 2018*). To construct CellPlex libraries, followed 10× Genomics Chromium 3' V3.1 and V3.1 HT kit with Cell Multiplexing. To construct targeted gene expression libraries for initial screen, amplified transcriptome libraries following the 10× Genomics library re-amplification protocol. Following re-amplification, libraries were pooled and targeted gene expression libraries constructed as in the 10× Genomics protocol. List of probes and primers used in *Supplementary file 1*. The genomic datasets presented here span 13 independent biological samples: the initial screen in *Figures 1 and 2* represents one biological sample, the focused screen in *Figures 3 and 4* represents another independent biological sample, the TBX5 enhancer deletion studies in *Figure 5* represent eight independent biological samples (four genotypes with two independently cultured biological replicates each), and *Figure 6* contains three more biological samples (three genotypes with one biological replicate each).

## Sequencing

Libraries were sequenced through a combination of Illumina NovaseqS4, NextSeq 500/550, and NextSeq 2000. We used paired-end sequencing using the following settings: R1-151bp R2-151bp idx1-10bp idx2-10bp on Novaseq; R1-28bp R2-54bp idx1-10bp and R1-28bp R2-56bp idx1-8bp on NextSeq 500/550; and R1-28bp R2-90bp idx1-10bp idx2-10bp on NextSeq 2000. All sequencing data is available on GEO: GSE190475.

## Mapping

For the initial screen and focus screens, scRNA-seq libraries were demultiplexed and mapped to the human reference genome (hg38) using the Cellranger software (ver 3.1.0, 10× Genomics). Focus screen experiments were mapped using Cellranger count with the following flags; `--expect-cells=10000`, and `--chemistry=SC3Pv3`. Initial screen experiments were mapped using Cellranger count with the following flags; `--feature-ref (cell hashing library)`, `--expect-cells=45000`, and `--chemistry=SC3Pv3`. Cell hashing antibody assignments were included as a feature barcode. scRNA-seq mapped libraries for each experiment were combined using Cellranger aggr and normalized for sequencing depth using the flag `--normalization=mapped`. Sequencing exp (*Figure 5*) was mapped to the human reference genome (hg38) using the Cellranger software (ver 6.0.0, 10× Genomics) with Cellranger multi using the following parameters; `--expect-cells=30,000` and `--chemistry SC3Pv3`. Sequencing exp (*Figure 6*) was mapped to the human reference genome (hg38) using Cellranger software (ver 6.1.2, 10× Genomics) with Cellranger multi using the following parameters: `--expect-cells=60,000` and `--chemistry SC3Pv3HT`. Cellplex oligo assignments were included as a feature barcode. Cellplex demultiplexed libraries were aggregated using

Cellranger aggr and normalized for sequencing depth using the flag `--normalization=mapped`. sgRNA libraries were mapped using FBA (*Duan et al., 2021*). FBA extract was run using a mismatch of 1 with `--r1_coords` 0,16 and `--r2_coords` 19,38. Following sgRNA sequence extraction, FBA count was run. Initial screen targeted gene expression libraries were mapped to the human reference genome (hg38) using Cellranger (ver 6.1.2, 10× Genomics) count with the following flags; `--expect-cells=45,000`, and `--chemistry=SC3Pv3`. Initial screen targeted gene expression libraries were aggregated using Cellranger aggr and normalized for sequencing depth using the flag `--normalization=mapped`.

## Single-cell processing

We assigned sgRNA to cells using the saturation curve method described in Drop-seq (*Macosko et al., 2015*). For a given sgRNA, we calculated the cumulative distribution of the UMIs from all cells. By identifying the inflection point of the curve, we adjusted the UMIs from cells after the inflection point to be zero. The sgRNAs with the adjusted UMI count greater than 0 are considered as true sgRNAs in this cell. To process cell hashing antibodies, we applied a saturation curve in a similar method as with the sgRNA processing.

We filtered for only singlets using Scrublet in the focused screen. In our initial screen, cells in which only one antibody barcody was detected were retained. Cells with higher than 20% mitochondrial content were removed, as well as genes with fewer than 1 counts. The count matrix was natural log transformed and saved as a separate layer for differential gene expression calling before scaling counts to unit variance and zero mean. An extra filter was applied to the initial screen in which cells where we could not detect an sgRNA were removed. Post filtering we were left with 80,147 cells in the initial screen. Clustering was performed through Scanpy implementations (*Wolf et al., 2018*). Briefly, PCA was performed followed by neighborhood calculation using default parameters. We then clustered cells using the Louvain algorithm at 0.2 resolution for initial and focus screens, 0.15 resolution for sequencing exp (*Figure 5*), and 0.1 resolution for sequencing exp (*Figure 6*). PAGA was performed to map cluster connectivity and served as the embedding for UMAP visualization, which was applied with default settings. Clusters which held fewer than 1% of all cells were removed. This resulted in 22,214 single cells in the focus screen, 31,367 cells in sequencing exp (*Figure 5*), and 24,571 in sequencing exp (*Figure 6*).

We applied the rank_gene_groups function of scanpy to identify cluster defining genes using the Wilcoxon rank-sum method. In our initial screen, we identified the following populations: two neuronal (+SOX2), CMs (+TNNT2), mesoderm (+FN1), epithelial (+EPCAM), cardiac fibroblast (+COL3A1), and endoderm (+TTR). In our focus screen, we grouped similar clusters together and defined the following cell types: neuronal (+SOX2), CMs (+TNNT2), endoderm (+TTR), and endothelial (+PECAM1). Similar clusters were observed in sequencing exp (*Figure 5* and *Figure 6*). For all experiments, we identified the enriched genes defining each Louvain cluster by applying the Scanpy rank_gene_groups function using the Wilcoxon rank-sums method.

## Single-cell trajectory clustering

To identify CM trajectory, clusters were filtered to retain cell populations relevant to cardiac differentiation. We isolated Louvain clusters corresponding to CMs in the focus screen and initial screen. The mesoderm cluster was included in the initial screen due to its involvement in CM differentiation. We then clustered these cells using PCA and PHATE with k=10/a=20 for the focus screen, k=30/a=100 in the initial screen, k=10/a=100 in sequencing exp (*Figure 5*), and k=10/a=50 in sequencing experiment (*Figure 6*). Neighborhood calculation and Louvain clustering at resolution 0.83 for the focus screen, 0.3 for the initial screen, 0.6 in sequencing experiment (*Figure 5*), and 0.7 in sequencing experiment (*Figure 6*) were applied to generate new clusters. We selected the clusters corresponding to CM and smooth muscle markers resulting in 1832 focus screen, 17,716 initial screen, 4717 sequencing experiment (*Figure 5*), and 4871 sequencing experiment (*Figure 6*) CM cells. The above clustering process was then repeated using k=10/a=20 for the focus screen, k=60/a=100 for the initial screen, k=10/a=100 in sequencing exp (*Figure 5*), and k=20/a=100 for sequencing experiment (*Figure 6*). We applied the Louvain algorithm to the initial screen, focus screen, and sequencing experiment (*Figure 5*) at resolution 0.3. For sequencing exp (*Figure 6*), an initial broad Louvain clustering was identified using resolution 1.2. This was used to identify the

root mesoderm-like cluster for pseudotime calling. Following, a final resolution of 0.4 was applied. Louvain clustering identified SOX4+ progenitor, FN1+ early CMs, ACTA2+ mid-CMs, and NPPA+ atrial-like late CMs in our initial screen. Our focus screen identified three clusters comprising progenitor, mid- and late CMs. Sequencing exp (*Figure 5*) consisted of three clusters comprising early, mid-, and late CMs. Sequencing exp (*Figure 6*) consisted of four clusters comprising early, mid-, late, and IRX2+ ventricular CMs. To verify the integrity of the PHATE projected trajectory, we ran diffusion pseudotime as implemented in Scanpy dpt. Clusters reminiscent of mesoderm-like cells were selected as the roots for all projections.

We applied the ingest module of Scanpy to transfer CM cluster labels across datasets as validation. We assigned the initial screen as reference and transferred labels onto focus screen CMs. Sequencing exp (*Figure 6*) was also assigned as reference and labels transferred onto other CM datasets. Perturbation CM distribution analysis was repeated across all label transfer re-assignments.

## sgRNA filtering

We designed multiple sgRNA for each candidate enhancer and promoter to account for potential off-targets. To identify sgRNA with off-target effects or that were non-functional, we applied a filtration approach using the identified CM trajectory in the large experiment. First we filtered our five non-target sgRNA to create a background. This was done by using a hypergeometric test for each CM cluster to compare the distribution of cells with each individual NT to the background of cells with any NT. We performed the hypergeometric test five times, in each iteration removing one of the NT. Through this we identified one NT which, if included, produced bias in the background distribution. After removing this NT, we were left with four similarly distributed controls which served to define our control cells for downstream analysis.

Second, for each target, we filtered to remove sgRNA that deviated in distribution to the majority. For each possible combination of sgRNA cells for each target, we performed a hypergeometric test for each CM cluster comparing the distribution of these sgRNA cells to the filtered NT cells background. For each of four CM clusters, we had two lists of p-values, one for depletion of sgRNA cells and one for enrichment. This resulted in a total of eight p-value lists. We then ordered each resulting p-value list for all sgRNA combinations in descending order of significance. A cutoff was drawn corresponding to the number of times each sgRNA is found in a given combination. We then asked whether any given sgRNA for this target was found at a significant rate to the left of this cutoff by using a hypergeometric test. If an sgRNA was found to be uniquely biased in this direction, it was removed. This was repeated for all eight p-value lists. If more than half of the sgRNA for a given target would be removed by this approach, all sgRNA were kept. This was repeated for sgRNA targeting every enhancer and promoter. All analysis, including validation experiment, used this filtered sgRNA set.

## Bulk validation of CM trajectory

Previously published RNA-seq data from an hESC to CM differentiation time course was mapped to the hg38 reference genome by STAR. For each sample, counts were normalized to RPKM. Replicate counts for each sample were then averaged together. We identified the top 100 uniquely enriched genes for each of our CM subpopulations in the initial screen from the Scanpy rank_gene_groups function using the Wilcoxon rank-sums method. For each enriched cluster gene set, fold change over day 0 was calculated for every time point. The fold changes of each gene from a gene set were then averaged by time point.

## Comparison of sgRNA distribution

Clustering bias was performed in a similar manner for every dataset. We compared the distribution of sgRNA for a given target to NT sgRNA across cell clusters using a hypergeometric test. Depletion of sgRNA was calculated using the hypergeometric cumulative distribution function and enrichment by using the survival function.

To compare the distribution of sgRNA across CM trajectory pseudotimes, we applied a Mann-Whitney U test which identified whether the median pseudotime for a given target was significantly shifted from NT control.

## Differential expression of cell type defining gene sets

We applied a Wilcoxon rank-sum test using Scanpy integration to compare gene expression of late CMs and mid-CMs from our focused screen. This approach identified 18 genes significantly enriched in mid-CMs and 80 genes enriched in late CMs. To assess the expression of these gene sets in NT control vs sgTBX5 ENH cells within the late-CM population, we calculated the average expression of this gene set within all our sgTBX5 ENH cells in late CM. As significantly more NT cells (558) achieved a late-CM fate in comparison to sgTBX5 ENH cells (122), we obtained a control expression distribution by sampling 122 NT cells through 1000 iterations and recording the average expression of genes sets each time. We applied a Z-test comparing perturbed to control expression and converted to p-value using scipy.stats.norm.cdf for enrichment and scipy.stats.norm.sf for depletion of gene set expression in perturbed cells. The same Z-test methodology was applied to our knockout screen in late CMs comparing focused screen differential gene sets between knockout and WT cells.

## Differential expression

Differential expression of individual genes was calculated by applying a Mann-Whitney U test through scipy.stats.mannwhitneyu to the expression distribution of a given gene in perturbed to NT control cells.

## Acknowledgements

We acknowledge the BioHPC computational infrastructure at UT Southwestern for providing HPC and storage resources that have contributed to the research results reported within this paper. GCH is supported by CPRIT (RP190451), NIH (DP2GM128203, UM1HG011996, 1R35GM145235), the Burroughs Wellcome Fund (1019804), the Welch Foundation (I-2103-20220331), and the Green Center for Reproductive Biology. NVM was supported by the NIH (HL136604, HL151650, and UM1HG011996), the Burroughs Wellcome Fund (1009838), and the Department of Defense (PR172060).

## Additional information

### Funding

| Funder | Grant reference number | Author |
| --- | --- | --- |
| NIH | DP2GM128203 | Gary C Hon |
| NIH | UM1HG011996 | Nikhil V Munshi<br>Gary C Hon |
| NIH | 1R35GM145235 | Gary C Hon |
| CPRIT | RP190451 | Gary C Hon |
| NIH | HL136604 | Nikhil V Munshi |
| NIH | HL151650 | Nikhil V Munshi |
| Burroughs Wellcome Fund | 1019804 | Gary C Hon |
| Burroughs Wellcome Fund | 1009838 | Nikhil V Munshi |
| Welch Foundation | I-2103-2022033 | Gary C Hon |
| Department of Defense | PR172060 | Nikhil V Munshi |

The funders had no role in study design, data collection and interpretation, or the decision to submit the work for publication.

### Author contributions

Daniel A Armendariz, Software, Formal analysis, Investigation, Visualization, Methodology, Writing - original draft; Sean C Goetsch, Formal analysis, Validation, Investigation, Methodology, Writing - original draft; Anjana Sundarrajan, Sushama Sivakumar, Validation, Investigation; Yihan Wang, Formal

analysis; Shiqi Xie, Conceptualization; Nikhil V Munshi, Gary C Hon, Conceptualization, Resources, Supervision, Funding acquisition, Project administration, Writing - review and editing

**Author ORCIDs**
Daniel A Armendariz ⓘ http://orcid.org/0000-0002-5857-1282
Sean C Goetsch ⓘ http://orcid.org/0000-0003-1141-683X
Sushama Sivakumar ⓘ http://orcid.org/0000-0001-7877-4821
Nikhil V Munshi ⓘ http://orcid.org/0000-0002-8397-100X
Gary C Hon ⓘ http://orcid.org/0000-0002-1615-0391

**Decision letter and Author response**
Author response https://doi.org/10.7554/eLife.86206.sa2

## Additional files

### Supplementary files
• Supplementary file 1. List of target enhancers and sequences used in study.
• Supplementary file 2. Sequencing statistics.
• Supplementary file 3. Top 100 Louvain cluster defining genes for each dataset.
• Supplementary file 4. Differentially expressed genes through hypergeometric test.
• Supplementary file 5. Focus screen mid-CM and late-CM defining genes.
• Supplementary file 6. Differentially expressed genes between TBX5 enhancer 5 heterozygous KO cells and WT cells, in the late-CM state.
• MDAR checklist

### Data availability
Sequencing data have been deposited in GEO under accession code: GSE190475. We have deposited Jupyter notebooks to GithHub (https://github.com/darmen04/Repression-of-CHD-associated-enhancers-delays-human-cardiomyocyte-lineage-commitment copy archived at *Armendariz, 2023*).

The following dataset was generated:

| Author(s) | Year | Dataset title | Dataset URL | Database and Identifier |
|---|---|---|---|---|
| Armendariz D, Hon G | 2022 | CHD-associated enhancers direct human cardiomyocyte lineage commitment | https://www.ncbi.nlm.nih.gov/geo/query/acc.cgi?acc=GSE190475 | NCBI Gene Expression Omnibus, GSE190475 |

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
