## [Editor Report]

The work presented is a valuable assessment of a broad set of regulatory elements that coordinate cardiac differentiation. The approach is broadly applicable, and the results point to important mechanisms of gene regulation and differentiation, with implications for future studies in non-coding variation.

---

## [Author Response]

[Editors' note: we include below the reviews that the authors received from another journal, along with the authors’ responses.]

Summary of changes

1. We thank the reviewers for their careful review of our manuscript and their helpful comments. We have made several key updates to the manuscript, including:

2. We generated and validated H9 hESCs with TBX5 exon deletion, and we performed comparative studies with TBX5 enhancer knockouts during cardiomyocyte differentiation.

3. We performed new experiments to independently phenotype TBX5 enhancer KOs in comparison to TBX5 exon KOs and WT cells. New experiments included: single-cell RNA-Seq, qPCR, immunocytochemistry, and FlowFISH (PrimeFlow).

4. We performed target amplification studies of putative enhancer target genes to verify gene expression changes.

5. To validate the integrity of cell cluster labeling, we performed cell label transfer analysis.

6. We improved the single-cell analysis to give more molecular insights on enhancer perturbations, including downstream target genes.

Below, we provide point-by-point responses to the comments. All the responses have been incorporated into the revised manuscript.

Reviewer #1It is a shame that the authors present broad cell state as a readout, rather than more in depth reporting of gene expression data.

We agree that more in-depth reporting of gene expression phenotypes would be helpful. To address this issue, we overcome several technical challenges.

– First, some cardiac genes are lowly expressed, making differential gene expression analysis underpowered. Hence, we have performed targeted sequencing to measure the expression of key cardiac gene targets (Figure 2—figure supplement 1D).

For example, this analysis shows that CRISPRi of TGFB1 promoter and enhancer results in 28% and 38% knockdown, respectively (Figure 2—figure supplement 1E).

– A second technical challenge is that CRISPRi repression results in incomplete silencing in the single-cell screens. For example, our positive control of targeting MALAT1 achieves 28% silencing (Figure 1G). To address this concern, we have generated additional enhancer knockouts, focusing on TBX5 regulatory elements. RNA sequencing of these full knockouts revealed gene expression signatures of fully penetrant enhancer perturbations. For example, knockout of the TBX5 exon resulted in repression of key cardiac development genes, which is phenocopied in TBX5 enhancer knockouts. These include expected targets of TBX5. See 3 Manhattan plots in Figure 5K, Figure 5—figure supplement 1D.

– A third technical challenge is that, by design, our cardiac differentiation strategy is only modestly efficient, and each perturbed element only has a modest number of cells attaining the mesodermal state. To address this issue, we combined cells with perturbed enhancers of the same gene. This allows us to pinpoint genes that were differentially expressed. See Manhattan plots in Figure 2O and Figure 4—figure supplement 1C.

(Initial Screen) Differentially expressed genes in CM cells with an sgRNA targeting a TBX5 enhancer. Please note the loss of NPPA and NPPB expression.

(Focused Screen) Differentially expressed genes in CM cells with an sgRNA targeting a TBX5 enhancer. Please note the loss of NPPA and NPPB expression.

Why would a TNNT2 enhancer inhibition alter cell state rather than just contractile function?

While TNNT2 is not known to function as a transcriptional regulator, recent results from similar screens have identified unexpected gene expression phenotypes for genes not typically associated with regulatory function. For example, Replogle et al. showed that knockdown of chromosome segregation genes (with unclear roles in transcription) resulted in gene expression phenotypes (Replogle et al. 2022). Similarly, genes with roles in glycolysis, vesicular trafficking, and DNA replication also unexpectedly caused transcriptional changes. These observations make possible the systematic assessment of genotype-phenotype relationships by using the transcriptome as a readout. Thus, TNNT2 knockdown may impact transcriptional states through similarly unexpected effects. One possibility is that, since TNNT2 is a critical component of the sarcomeric apparatus, its loss could initiate a mechanosensory feedback response resulting in delayed CM specification. In support of this possibility, we observe that CRISPRi perturbations of two TNNT2 enhancers and the promoter consistently yield changes in transcriptional and cell state phenotypes (Figure 2G) (Figure 2I).

How do the data for the TBX5 enhancers compare with the gene knockout data?

To adequately address this reviewer’s concern, we have generated and extensively validated TBX5 exon knockouts in H9 hESCs. Overall, we observe that:

– Knockout of TBX5 enhancers partially phenocopies TBX5 gene knockout. We find that TBX5 gene and enhancer knockouts exhibit reduced TBX5 protein expression by immunocytochemistry (Figure 5C-E).

– Consistently, the TBX5 downstream target NPPA also exhibits reduced protein expression by immunocytochemistry (Figure 5D).

–To examine changes in cell state, we performed new single-cell RNA-Seq experiments in all exon and enhancer 3 / 5 knockouts. We observe that TBX5 gene and enhancer knockouts also result in deficient differentiation of CMs (Figure 5I-J and Figure 5—figure supplement 1D).

–Consistent with an increase in earlier stage CMs, we find that TBX5 gene and enhancer knockouts have increased RNA expression of FN1 Figure 5K-N shows data from the single-cell RNA-Seq experiments.

– qPCR and western blot experiments also confirm this result (Figure 5L and 5N)

–To test if we can observe increased FN1 expression using an orthogonal assay, we performed FlowFISH experiments, which uses RNA in situ hybridization to examine expression in individual cells. Consistent with an increase in earlier stage CMs, we find that TBX5 gene and enhancer knockouts have increased RNA expression of FN1 (Figure 5M).

What are the type of CHDs represented in this set of loci?

CHD variants proximal to our target loci encompass a diverse set of CHDs including atrial and septal defects and Tetralogy of Fallot. We have added a table with patient phenotype information (Supplementary file 1).

The analysis falls short of bringing truly new insight into CHDs, cardiac differentiation, or the specific function of enhancers in these processes. As the iPS cell system is 2D, and apparently in the authors' hands very (surprisingly: neurons?) heterogeneous in its output, the relevance and importance of the findings as presented is limited.

Yes, we agree that standard cardiac differentiation strategies are less heterogeneous, but they use highly stringent metabolic selection to enrich for cardiomyocytes. We sought to avoid this selection because it could mask subtle phenotypes resulting from promoter and enhancer perturbations. For example, mice lacking TBX5 do not produce hearts (Bruneau et al. 2001), but ESCs lacking TBX5 can still differentiate into cardiomyocytes (Zhao et al. 2021). Therefore, we eliminated the metabolic selection step during hESC differentiation, resulting in the production of heterogeneous cell states including both mesodermal (cardiomyocytes and progenitors), ectodermal (neuronal), and endodermal states. This strategy was previously used by Tohyama et al. (Tohyama et al. 2013).

In support of the relevance of this system, Zic2 is required for early specification of cardiomyocytes. In our system, CRISPRi knockdown of Zic2 results in loss of cardiomyocyte and neuronal specification. These cells with Zic2 sgRNAs are almost entirely diverted to the mesodermal lineage, and yet do not efficiently differentiate into cardiomyocytes (Figure 1—figure supplement 1H). These results suggest that the system is relevant and that the results from other perturbations are also potentially relevant to CHD.

The authors state that loss of TBX5 alters CM lineage. In fact the reduced dosage of TBX5 did not alter this, and complete loss of TBX5 only altered later differentiation endpoints. So the authors' partial (slightly over 50%) reduction in TBX5 is in fact consistent with the allelic deletion series.The conclusion that "these enhancers have roles in CM lineage commitment." Is not consistent with the data, which show (incompletely) modestly reduced numbers of late CMs. This would be a decreased differentiation efficiency.

We agree with the reviewer. Work from Kathiriya et al. identified that “TBX5in/+ cells followed a path similar to WT but to a transcriptionally distinct endpoint”. As correctly pointed out by the reviewer, our TBX5 enhancer knockdown results are quite consistent with the allelic series of the prior study. Both TBX5 hypomorphs and enhancer knockouts display similar populations with inefficient cardiomyocyte differentiation. We have modified the text to reflect this observation.

We agree with the reviewer on the subject of accurate language. We have modified the text to reflect a decrease in differentiation efficiency. An example is shown below:

Before: “We showed that loss of cardiac enhancer function results in delayed CM specification…

After: “We observe that perturbation of CHD-associated enhancers, particularly for TBX5, results in deficient CM differentiation.”

Reviewer #2Major concerns1. The efficiency of CRISPRi knockdown was only validated for the promoters MALAT1 and OCT4, however there is no evidence of enhancer loss-of-function provided. Considering all findings from the study rely on the robustness of the initial large-scale CRISPRi screen, it is critical the authors provide clear evidence on the efficacy of this system in repressing target enhancer activity. While the functional validation of knockdown efficiency for all sgRNAs is unreasonable, at a minimum the loss of enhancer activity should be confirmed via qPCR or CAGE-seq. We recognise this is a very challenging QC assay as it relates to validating knockdown of enhancer elements (vs genes) – but it is essential to have this to ensure fidelity of downstream data interpretation and assays.

We agree that we did not provide direct evidence of enhancer loss of function. We have performed two experiments to mitigate this.

First, we have performed TBX5 qPCR of D8 CMs following transduction with either non-target sgRNA or sgRNA targeting TBX5 enhancer 3 or 5. We observe marked reduction of TBX5 when either enhancer is repressed (Figure 1—figure supplement 1I).

Second, we performed target transcript amplification of 25 cardiac genes proximal to our targeted enhancers. This allowed for the robust detection of lowly expressed genes such as TGFB1 where we can detect repression in either TGFB1 promoter or enhancer knockdown cells (Figure 2—figure supplement 1E).

2. The 'NC control' referred to throughout the manuscript is not defined. This is critical, as this control is used as a reference point in all analyses included in this study. Are these cells lentivirus treated cells without gRNA or are these non-treated cells? The inclusion of an appropriate control is crucial, especially considering studies have shown that the introduction of lentiviral vectors in cells induces a robust innate immune response and technical factors such as these could influence the differentiation efficiency.

We agree that non-treated cells are an inappropriate control. In our studies, we use negative control sgRNAs that are lentivirally transduced together with targeting sgRNAs. Thus, negative controls sgRNAs are an internal experimental control in the pooled CRISPR screen.

We apologize for the confusion in our notation. In the text, we used “NC” to refer to a set of “negative control” sgRNAs, which do not target the genome: either targeting GFP, the CAG Promoter, or empty sgRNA. We now explicitly define this in the text. To further reduce confusion, we use “NT” to refer to these “non-target” sgRNA controls.

3. The annotation of cell types in all scRNA-seq datasets are poorly justified. Rather than relying on individual gene markers for manual annotation, we suggest the use of label transfer or other robust methods for unbiased cell type annotation. This is a major concern considering the underlying claims made in this study rely heavily on the quality of this initial cell type labeling step. This is relevant to all single cell analyses performed in the study.

We thank the reviewer for the suggestion. To address this concern, we first defined CM subpopulations for the initial screen. This was done using a combination of marker genes and pseudotime ordering of clusters (Figure 2A) (Figure 6F). We then applied the ingest function within the scanpy python package to label CMs in the focused screen using the initial screen as reference (Figure 4—figure supplement 1D). Cell label transfer produces similar clustering to our manual annotations and also supports the observation of deficient CM differentiation in hits (Figure 4—figure supplement 1E).

Similar analysis using the sequencing exp (Figure 6) datasets as a reference also yield consistent results (Figure 6—figure supplement 1H-K).

4. The consistent reliance on the proportional distribution of cells in scRNA-seq data (Figures 1H, 2G, 2J, 2L, 3E, 4F, 5E, 5H) is not sufficient to justify the claim that loss of target enhancers disrupts the proper differentiation of CMs as the authors claim. In some cases, the differences in proportions are so small (e.g. Figure 1H) that their biological significance is difficult to interpret. These data are correlative and require validation via independent methods such as qPCR, FACs, imaging or other independent endpoints to support claims about effects of enhancer KD on differentiation.

To address the reviewer’s concern, we have performed several independent assays to assess CM phenotypes using several genetic knockouts: TBX5 exon3 -/-, TBX5 enhancer 3 -/-, and TBX5 enhancer 5 -/-.

– qPCR: We performed qPCR for NPPA (late CM marker), and FN1 (early CM marker). In all knockouts, we observed elevated levels of FN1 and depletion of NPPA (Figure 5C and 5L).

– FlowFISH: To test if we can observe increased FN1 expression using an orthogonal assay, we performed FlowFISH experiments, which uses RNA in situ hybridization to examine expression in individual cells. We performed FlowFISH for FN1 in all exon and enhancer knockouts. Briefly, cells are stained with fluorescent probes targeting FN1 and RPL13A control transcripts. After staining, FN1 expression is quantified at the single cell level through flow cytometry, relative to the RPL13A control. We observe elevated levels of FN1 transcript through PrimeFlow (Figure 5M).

– Western Blot: As an alternative validation of increased FN1 expression, we examined protein expression of FN1 across our TBX5 genetic knockouts using western blots. Western blot confirmed elevated FN1 across the knockouts (Figure 5N).

– ICC: As a further validation, we examined protein expression of TBX5 and NPPA across our TBX5 genetic knockouts using immunocytochemistry. ICC phenocopied scRNA-seq data, showing decreased TBX5 and NPPA across the knockouts (Figure 5D-E).

Overall, these results support the observations from single-cell analysis that TBX5 gene and enhancer perturbations cause deficient CM specification.

5. Cellular phenotyping of TBX5-enhancer CRISPR knockout cells via flow cytometry, protein expression and calcium transient measurements displayed no significant difference to control (Figure S5), deriving >90% TNNT2+ cardiomyocytes on day 23 of differentiation. These results are discordant with the findings from the scRNA-seq analysis of day 8 TBX5-enhancer knockout cells where the authors claim effects of TBX5 enhancers on cardiomyocyte specification. We suggest the authors perform secondary validation of perturbed cells at day 8 (e.g. using FACs and qPCR) to assess the effects of TBX5-enhancer knockdown on the cardiac lineage.

Following the reviewer’s suggestion, we performed a panel of secondary validation experiments, including qPCR, FlowFISH, and immunocytochemistry, on TBX5 exon and enhancer genetic knockouts: TBX5 exon3 -/-, TBX5 enhancer 3 -/-, and TBX5 enhancer 5 -/-. These analyses were conducted at day 8 of differentiation to match our scRNA-seq analysis, and are also mentioned above in Response 4. Consistently, these data support the molecular phenotypes observed in the single-cell data and are also in agreement with the conclusions from the study by Kathiriya et al.

We agree with the reviewer that comparing cellular phenotypes at day 23 of differentiation (when these assays can be readily performed) to single-cell RNA-Seq data at day 8 is not ideal. To avoid confusion, therefore, we have removed day 23 data and focused on day 8.

6. Figure 5H: these data point to differences between ENH3-/- and ENH5-/- groups suggesting that these enhancers may be acting differently to influence transcriptional programs during cardiac differentiation. However, these differences are not addressed in the text and the authors claim that they have the same role. These difference are perhaps the most interesting in the manuscript and could be further developed because this may provide insight into the regulatory function of different enhancers in controlling cardiac lineage specification. Can the authors develop this finding further as a key area of novelty and significance in the study?

Response 6: During revision, we determined that the TBX5 ENH5 clone described in the first submission was a functional heterozygous deletion (rather than a homozygous deletion) (Figure 6B). We have since rederived and confirmed TBX5 ENH5 -/- hESC clones. After repeating the CM differentiation and single-cell sequencing, we observe minimal differences between ENH3 and ENH5 KOs in regards to differentiation deficiency (Figure 5I). We have revised the section regarding Figure 5 accordingly. In addition, we have added a new paragraph describing the result of the heterozygous ENH5 +/- (derived in the first version of this manuscript). We conclude that ENH5 +/- cells do not display differentiation deficiencies to the degree of full enhancer knockouts. These observations are in-line with those observed for the TBX5 gene deletions previously published.

Minor concerns1. Figure 2E: The order in which the targets are displayed is confusing and makes it difficult to interpret. It is suggested the authors display the enhancer targets in a separate panel to the promoter targets.

We thank the reviewer for the suggestion and have updated the figure (Figure 2E) jnto three panels to display promoters, enhancer, and controls.

2. It is unclear what additional novel insights are gained from the secondary CRISPRi screen of 6 TBX5 enhancers- the data appear to confirm the same conclusions (e.g. Figures 3E-F and 4F-G) already drawn from the inclusion of these targets in the large-scale screen (Figure 2L-M). Can the authors clarify this point?

The main purpose of the secondary CRISPRi screen is to confirm the observations from the initial screen given our concern for potential false positive hits. Not only did we validate the primary screen, but we also found that the secondary CRISPRi screen exhibited greater repression as portrayed by a general depletion of TBX5 enhancer knockdown cells within the CM cluster (Figure 4F). Overall, the data from the secondary screen phenocopied what was observed in the primary large-scale screen and allowed for differential expression analysis since CRISPRi repression was stronger (Figure 4H-O).

3. Throughout the manuscript the authors use jargon and vague language that undermines the narrative of the study. For example: "delays CM fate commitment by altering transcriptional signatures", "results in delayed transcriptional specification of CMs." The authors should carefully review the manuscript to ensure accuracy of results and interpretation.

We thank the reviewer for their comment. We have opted for more precise language which has been corrected in the text. An example of modified language is below:

Before: Taken together with previous observations, misregulation of TBX5 through enhancer modulation delays CM fate commitment by altering transcriptional signatures.

After: Taken together with previous observations, misregulation of TBX5 through enhancer modulation leads to deficient induction of CM transcriptional signatures that are associated with CM fate commitment.